# The Impact of Consumer Loyalty and Customer Satisfaction in the New Agricultural Value Chain

Chin-Shyang Shyu [1], Chun-Chang Yen [2,*] and Cheng-Sheng Lin [3,*]

1   Department of Recreation and Holistic Wellness, MingDao University, Changhua 52345, Taiwan; shaw@mdu.edu.tw
2   Department of Sports and Leisure Management, Meiho University, Pingtung 91200, Taiwan
3   Department of Agricultural Technology, National Formosa University, Yunlin 63201, Taiwan
*   Correspondence: x00008464@meiho.edu.tw (C.-C.Y.); sheng8876@nfu.edu.tw (C.-S.L.)

**Abstract:** Affected by advances in artificial intelligence and awareness of green environmentalism, consumers' purchasing behavior of fresh agricultural products has been transformed. New agricultural value chains are being created, such as the new retail model. This study used principal component analysis (PCA) to select essential factors of online consumer behavior and adopted multilevel structural equation modeling (MSEM) to analyze the consumer behavior of Fresh Hema, a new retail format completely reconstructed by Alibaba for offline supermarkets, under different levels of the new retail model. The study adopted an online questionnaire of the consumer behavior of Fresh Hema in 2022. The results show that playfulness and epidemic prevention positively impacted customer satisfaction; convenience benefits and green logistics had the most significant positive impact on customer loyalty.

**Keywords:** consumer loyalty; new retail; customer satisfaction

## 1. Introduction

The outbreak of COVID-19 in 2019 has affected the global economy, leading to high uncertainties across all industries. Barbaglia et al. [1] highlighted that economic shock represents an unexpected, unprecedented reaction of the economy to the changes, where no past observations could provide a relevant signal about its potential economic impact. During the COVID-19 epidemic, the traditional retail industry has been affected by people's reduced offline consumption. This has also influenced the sale of agricultural products and specialty foods. Petty farmers who maintain the rich diversity and ecology of the land through intensive cultivation also face a sales dilemma. Yang, Chen, and Liang [2] suggested that the global traditional retail industry faces unprecedented challenges. Business performance is seriously deteriorating in this confusing period of transformation and upgrades. Traditional retail enterprises are looking to adopt pragmatic measures, and how to upgrade this industry with the agricultural value chain has become a new focus for managers. Ganeshkumar and Khan [3] pointed out that the agricultural value chain has majorly divided into six actors: farm inputs, land and farming, storage and trading, agriculture processing, distribution and retail, and consumers. In this paper, we focused on distribution, retail, and consumers.

With the rise of Industry 4.0 and the advances in digital technology, people's consumption patterns have also changed. From the perspective of consumers, with the advent of the new retail era, retail online official websites and offline entity channels have connected to form O2O (online to offline; online and offline integration). This can provide consumers with different purchasing experiences. Wei [4] found that with the rise of high-end technologies such as artificial intelligence and big data, new consumption patterns are constantly emerging, from offline retail to online e-commerce, leading to a new retail mode that combines online e-commerce, offline shopping, and logistics. Sun et al. [5] found that this new

retail mode involves companies that rely on the internet using advanced technologies, such as big data and artificial intelligence, to improve and change their products' production, distribution, and sales processes. The above literature indicates that new retail provides many enterprises with new operation and marketing ideas. In other words, with the development of mobile applications and big data-related technologies, people's consumption habits have shifted from offline (physical stores) to online (online stores). Zhang [6] suggested that "new retail" is dominated by information technology (big data, internet, artificial intelligence, etc.), which can improve customer experiences (shopping scenarios that meet the different needs of consumers). In other words, the possibility that O2O's innovative thinking helps enterprises explore business opportunities has dramatically increased. From the perspective of enterprises, industries have also begun to develop and sell products based on a "consumer demand-oriented" approach. Typically, there are three implementation steps. First, companies must understand the problems people encounter when using or purchasing products or services. Second, companies must use smart technology and big data analysis tools to explore consumers' behavior habits, purchase motives, interests and preferences, satisfaction, and loyalty. Third, according to the results of consumer behavior analysis, they must adjust product design and marketing strategies. Furthermore, to cope with the advances in smart technology, stores should establish "full channel thinking", face the impact of online word-of-mouth brand promotion, and understand how to integrate offline and online resources. Utilizing the internet's advantages eliminates distance and brings more business opportunities for enterprises. Further, it helps establish a good brand image and maintain stable customer relations. Therefore, O2O's innovative thinking helps enterprises grow and keep pace with the times.

Recently, different industries have explored the application of the new retail concept. Wei [4] found that the "new retail" concept is widely discussed as a new thing introduced into the academic field by the business environment. Most of the relevant literature is theoretical analysis and case elaboration, whereas no articles have verified the necessity of the new retail model through empirical research. Therefore, the present study conducted empirical research to explore consumer behavior with Fresh Hema under the new retail model, based on smart technology. Fresh Hema has 6000 to 8000 SKUs (stock keeping units). Fresh products are its primary business, and the food sold in offline stores accounts for 80%. The company also constructed "Hema Village", which provides exclusive agricultural products for Fresh Hema, according to actual order demand. With technology provided by Ali Digital Agriculture, cooperative farmers in each village can properly conduct field management according to their natural geographical conditions, resource advantages, and plant and produce-featured agricultural products. Agricultural products are gathered from scattered producers to the gathering yards for graded packaging and inspection through access to Hema's supply chain network and sales channels throughout China. Next, they are connected to the Hema new retail supermarket sales outlets and online stores. Therefore, traditional agriculture has been upgraded to digital agriculture, and traditional farmers are being transformed into agricultural industry workers [7].

According to the Report on China's Fresh Food E-commerce Market Data in H1 2023, the scale of fresh food e-commerce transactions in China is expected to reach RMB 642.76 billion in 2023. Additionally, in H1 2023, the fresh food e-commerce transactions of urban residents in China accounted for 27.6% of urban residents' food consumption expenditure. The major e-commerce companies in Shanghai include Dingdong Maicai, Jingdong Daojia, RT-Mart Youxian, Duoduo Maicai, Fruitday, and Hema Taiwan Trade Center in Shanghai [8]. Fresh Hema currently coexists with competitors using multiple business models. Fresh quality rather than price is the key factor in first- and second-tier cities in China. Thus, how to provide a better shopping experience for in-store consumption or online shopping through big data while achieving instant delivery to improve performance efficiency is a top priority for the new agricultural value chain. This is what consumers are truly concerned about. As such, the purpose of this study was to explore the correlation between Fresh Hema's value-creating strategy and consumer satisfaction. On the other

hand, the dividend of internet traffic is gradually disappearing, and the cost of website traffic is increasing. Fresh Hema creates the value of its products and services through Hema physical stores and catering services, enabling consumers to place orders through online apps and consume offline. However, with many alternatives for fresh agricultural products, if branding is low, once consumers become dissatisfied, such as by receiving stale products, they may turn to other e-commerce merchants for consumption. Therefore, examining consumer loyalty to Fresh Hema products was also a focus of this study.

The present study adopted a principal component analysis to select factors affecting online consumer behavior. Structural equation modeling (SEM) was used to construct a model of the consumer behavior variables of Fresh Hema under the new retail model based on smart technology. This paper also explores online fresh food shopping malls and petty farmers' production and marketing strategies under the latest retail trends.

## 2. Literature Review

### 2.1. Consumer Behavior in the Context of Agricultural Value Chains and Artificial Intelligence

Yan, Chen, Cai, and Guan [9] reported that the fresh agricultural product supply chain's circulation efficiency is significantly influenced by the purchasing power of end consumers. Consumers are an essential factor in determining agricultural value chains. Mowat and Collins [10] found that supply chains in new and emerging agricultural industries typically lack the ability to link product quality with consumer behavior.

Agricultural value chains have also changed over time. Dong [11] found that agricultural value chains are going through a paradigm shift from efficiency-driven to resilience-focused eco-friendly agriculture. Other factors affecting agricultural value chains, indicated by AI. Ganeshkumar, Jena, Sivakumar, and Nambirajan [12], include that, with AI adoption in agriculture, value chains can increase agricultural income, enhance competitiveness, and reduce costs. Among the agriculture value chain stages, AI research related to agricultural processing and the consumer sector is limited compared to that related to input, production, and quality testing. Based on the above relevant literature, it is evident that consumer behavior plays an essential role in agricultural value chains. The present study focuses on the increasing trend of consumers' demand for fresh agricultural products after COVID-19, addressing the gap in previous studies that primarily focused on general agricultural products [13–15]. This study also examines which parties are best suited to facilitate value chain upgrading.

### 2.2. Purchase Intention

Consumer behavior has been a focus of the retail industry for decades. Currently, while there is fierce competition among online retailers, more services and products are being added to online businesses to increase shopping convenience for customers. Analyzing consumer behavior is a significant factor for online business success. Online shopping is currently experiencing flourishing economic growth, which is also regarded as an associated benefit of e-commerce. The rapid development of this fundamental business idea has attracted consumers and vendors worldwide [16]. In the global environment affected by COVID-19 in 2020, consumer behavior has changed, due to consumers' inability to go to physical stores during the pandemic. This means that the COVID-19 pandemic has fueled the growth of the stay-at-home economy and zero-contact applications, such as distance work. The transfer of the consumption field has derived the transformation of the consumption pattern of "new commodity demand", "new consumption scenario", and "new consumption habits". Moreover, various retail industries have invested more resources in e-commerce; i.e., technology is being improved to strengthen purchase intention in the new retail era. The intention to purchase represents what consumers believe they will buy to satisfy their needs and wants in the future [17]. Consumers tend to buy products with higher perceived value; the higher their purchase intention, the higher the purchase probability. Montano and Kasprzyk [18] highlighted that intentions are considered the critical predictor of purchase intention actual behavior. Based on relevant studies, several factors

affect purchase intention. Dash, Kiefer, and Paul [19] explored the evolution of Marketing 4.0 and empirically examined its impact on customer satisfaction and purchase intention. Their study found that the impact of customer satisfaction on purchase intention is highly significant. Dhingra, Gupta, and Bhatt [20] analyzed the impact of the online service quality of e-commerce websites on customer satisfaction and purchase intention. The relationship between overall customer satisfaction and purchase intention was statistically significant. Mainardes and Cardoso [21] evaluated the effect of social media on consumer trust, loyalty, and purchase intention in physical stores. Their results indicate that loyalty positively impacts consumer purchase intention. Savila, Wathoni, and Santoso [22] conducted a quantitative study with structural equation modeling (SEM) using 311 respondent data from O2O e-commerce users in the Greater Jakarta area. Their findings show that offline customer loyalty drives customer repurchase intention.

Based on the results of previous studies, the present study posits the following hypotheses:

**H1:** *Customer satisfaction positively affects consumer purchase intention.*

**H2:** *Customer loyalty positively affects consumer purchase intention.*

*2.3. Customer Satisfaction*

Customer satisfaction is a post-purchase evaluation where the chosen alternative at least gives the same result or exceeds consumer expectations. At the same time, dissatisfaction would arise if the results obtained were not in accordance with or were below consumer expectations [23]. In other words, the overall attitude of customers after consumption can show their satisfaction with a product or service. Since enhancing customer satisfaction is an important strategy for changing consumer behavior and building consumer loyalty, it has become the focus of the industry. Ivana et al. [24] observed the influence of website quality on customer satisfaction and buying intention. Their results indicate that system quality positively affected customer satisfaction, and customer satisfaction positively affected purchase intention. Al Mulhem [25] investigated the effects of quality and organizational factors on university students' satisfaction with e-learning system quality. Their results indicate that quality factors (course content quality, system quality, and service quality) significantly positively affected students' satisfaction with e-learning system quality.

Rezaei and Valaei [26] investigated the experience of consumers shopping on smartphones and classified experiential value into four factors: CROI, service excellence, aesthetics, and playfulness, to determine their significant relationship with satisfaction. Playfulness has been found to be a significant antecedent that improves customer satisfaction. Davis and Boone [27] found that playfulness positively influences a person's attitude toward new systems and learning. Moon and Kim [28] found that perceived playfulness is a crucial factor affecting behavioral intentions. Ding, Feng, and Jiang [29] constructed an integrated framework to explore the direct and indirect relationships between four constructs: regular service quality, pandemic prevention service, psychological distance, and safety perception; they further examined passengers' satisfaction in the context of urban rail transit services. Their results indicate that pandemic prevention measures positively affected passenger satisfaction. Dhingra, Gupta, and Bhatt [20] analyzed the impact of the online service quality of e-commerce websites on customer satisfaction and purchase intention. They found that the relationship between overall e-service quality and customer satisfaction was statistically significant. Based on the results of previous studies, this study proposes the following hypotheses:

**H4:** *System quality positively affects customer satisfaction.*

**H5:** *Playfulness positively affects customer satisfaction.*

**H6:** *Pandemic prevention service positively affects customer satisfaction.*

**H7:** *Service quality positively affects customer satisfaction.*

*2.4. Customer Loyalty*

Customer loyalty refers to the continuous emotional relationship between a company or organization and its customers, which is manifested in the customers' repeated purchasing and continuous interaction with the company and is an integral part of the company's growth and development. Guarda, Villao, and Leon [30] suggested that customer loyalty is targeted through marketing strategies and tactics to ensure consumer value for the customer, so companies can retain them as loyal consumers. Gonçalves and Sampaio [31] found that the significance of customer loyalty is closely related to profitable growth. In the current e-commerce and new retail era, customer loyalty plays a more critical role in the long-term development goals of enterprises. Bhaskar and Kumar [32] found similar results, suggesting customer loyalty is essential to a company's long-term success and profitability. Furthermore, Guo, Zhang, and Xia [33] found that in a purchase situation, customer satisfaction and loyalty are primarily determined by usability, trust, and web design. Most existing research on customer loyalty views it as being related to expected behavior, specifically, purchase behavior on shopping websites. According to the findings of relevant studies, many factors affect customer loyalty. Santi, Sutomo and Zahara [34] aimed to determine the influence of experiential marketing on customer loyalty, with customer satisfaction as a mediating variable, with regard to the Bora Hot Springs tourist attraction. Their findings indicate that customer satisfaction significantly positively affected customer loyalty. Gunawan [35] examined matters relating to the study's title, namely the effects of experiential marketing and product quality on customer satisfaction and their impact on the customer loyalty of Uniqlo consumers in South Jakarta. Their results suggest that the best path analysis model for increasing customer loyalty is to focus on improving customer satisfaction. Widodo and Balqiah [36] studied the influence of PUBG Mobile addiction and perceived value, which consisted of playfulness and cost with regard to loyalty to PUBG Mobile.

The results showed there is a positive influence of playfulness on loyalty. Kaura, Prasad, and Sourabh [37] found that service convenience dimensions positively impact customer satisfaction and loyalty. To deliver information about products and services to customers, it is necessary to have a stable system so that customers are willing to use the service or system for a longer period. Nurdin and Abidin [38] aimed to determine what factors affect customer loyalty to Shopee e-commerce and tested how much influence the quality of Shopee's e-commerce recommendation system has on customer loyalty. Their results indicate that recommendation system quality in e-commerce could directly affect customer loyalty.

Online shopping has become convenient for customers worldwide. However, it negatively impacts the environment due to excessive packaging and materials used. Therefore, green logistics are crucial in the new retailer era. Kawa and Pierański [39] examined the main logistics challenges related to eco-friendly e-commerce to determine the impact of the green logistics approach in e-commerce on customer satisfaction and loyalty. Their empirical study confirmed the relationship between green logistics, satisfaction, and loyalty. Based on the results of previous studies, the present study posits the following hypotheses:

**H3:** *Customer satisfaction positively affects customer loyalty.*

**H8:** *Playfulness positively affects customer loyalty.*

**H9:** *Service convenience positively affects customer loyalty.*

**H10:** *Green logistics positively affects customer loyalty.*

**H11:** *System quality positively affects customer loyalty.*

## 3. Research Method

*3.1. Research Design*

This study used principal component analysis to select essential factors of online consumer behavior, and comprehensive, multilevel variables were integrated. Multilevel

structural equation modeling (MSEM) was adopted to analyze the relationships between the variables of Fresh Hema's consumer behavior at different levels of the new retail model. This paper also discusses the causality through a path map and determines the best marketing strategy for online fresh food shopping malls under the new retail model. As shown in Figure 1.

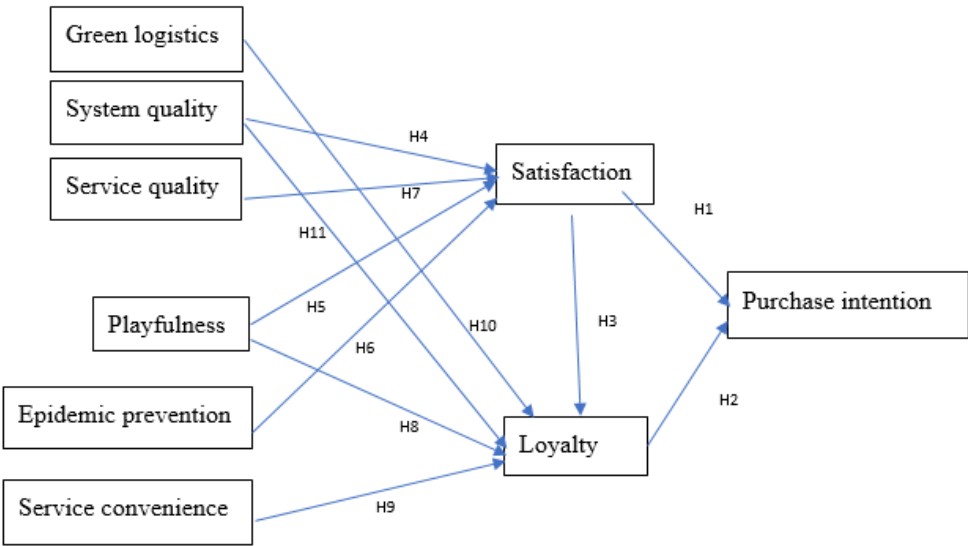

**Figure 1.** Research Model Framework.

The present study adopted MSEM. The main difference between single- and multi-level data analyses is that a multi-level data structure divides low-level individuals into different individual population variables based on a specific group variable and analyzes inter- and intra-group variation depending on the variable (Y). Muthen [40,41] noted that in a multi-level data structure, observation data at the global level are represented by the average of individual levels. Hence, estimates of inter-group variables contain information about intra-group variables. Currently, the S PW matrix is the most probable estimate of the parent matrix ($\sum W$). However, the $S`_B$ matrix is not the most probable estimate of the inter-group matrix ($\Sigma B$); the linear integrated weighted the most probable estimate of the inter-group ($\Sigma B$) and the in-group parent matrix ($\Sigma W$) as follows:

$$S_{PW} = \sum W \tag{1}$$

$$S`_B = \sum W + C_g \sum B \tag{2}$$

In multi-level data, individual and overall level observation units differ. The number of people in each group in Equation (2) is used for weighting. When the number of people in each group is the same, $C_g$ is a fixed constant called the balanced model. If the number of people in each group is not balanced (un-balanced model), $C_g$ is the variable. Muthén [40,41] suggested that the influence of the difference in the number of people in each group can be ignored. The $C_g$ weight can be replaced with the post hoc estimated group number ($C^*$) similar to the average number of groups to determine the observation matrix between groups. This achieves the purpose of model convergence by simplifying the model. The value of $C^*$ is defined as follows:

$$C^* = \frac{N^2 - \sum_{g=1}^{G} C_g^2}{N(G-1)} \tag{3}$$

where $N$ is the total number of samples and G is the number of groups. Once Equation (3) is used for maximum likelihood estimation, the solution is not an FIML solution, but the limited information maximum likelihood (LIML). Muthén [40,41] called this the MUML

(MUthén's ML) solution, whereas McDonald [42] referred to it as a pseudo-balanced solution. When the number of samples between and within groups is large, $S\grave{}_B$ is an estimate very close to $\Sigma B$. However, when the sample size is small, the deviation becomes more serious, and the parameter estimates and standard errors become more incorrect. Therefore, MSEM analysis adopts a two-stage procedure proposed by Anderson and Gerbing [43] to determine the most appropriate measurement model, and then proceeds to the structural model analysis. With potential variables (η) in the model, the inter- and intra-group structural models are defined as follows:

$$\eta_B = \alpha_B + B_B\eta_B + \varepsilon_B \tag{4}$$

$$\eta_W = B_W\eta_W + \varepsilon_W \tag{5}$$

Equations (4) and (5) define the matrix relationships between and within groups with potential variables. They are the primary form of multi-level structural equation modeling.

Due to the different locations of Fresh Hema stores and the extent of COVID-19, a combination of online and physical customer services under the new retail model also affects consumers' purchase motivation and behavior. It also indirectly affects consumers' satisfaction, loyalty, and purchase intention. During the epidemic, quarantine measures were implemented in various regions, resulting in multilevel and clustered consumer behavior data. If the traditional SEM model is used, the statistical observations will be invalid due to special dependence or duality, which will cause a violation of the hypothesis of sample independence and statistical tests. Therefore, this study used MSEM to analyze the influence of consumers' purchase motivation behavior on consumer satisfaction, loyalty, and purchase intention under the new retail model.

In this study, 70 Fresh Hema stores in Shanghai were taken as the basic clusters, and Multilevel Structural Equation Modeling (MSEM) was used to solve the above statistical error problem [44–48]. Furthermore, the consumption behavior at the individual level, as well as the impact of latent variables on the overall level of latent dependent variables was correctly evaluated. This was performed to measure the customer services and functions that best meet the needs of consumers to effectively improve the value supply chain of fresh agricultural products.

*3.2. Empirical Analysis*

This study's data are derived from online big data relating to consumer behavior of Fresh Hema in Mainland China in 2022 [49]. This study analyzed the transformation of consumers' purchasing behavior due to the impact of COVID-19, environmental protection and green energy. It explored the variable relationship between consumers' purchase motives, satisfaction, loyalty, and purchase intention, and it puts forward the best online marketing and management decisions.

This study extracted nine dimension variables using factor analysis. The reliability level of each dimension variable (Cronbach's alpha) was above 0.8. An SEM analysis was conducted based on the nine dimension variables, as shown in Table 1:

**Table 1.** Factor Analysis of the Dimension Variables.

| Dimension Variable | Explain Variables and Descriptions |
|---|---|
| 1. Green Logistics | G1: Use environmentally friendly packaging. |
| | G2: Use environmentally friendly shopping bags. |
| 2. System Quality | SQ1: Combine with network video system |
| | SQ2: Provide all product information on the network. |
| | SQ3: APP mobile payment service. |
| 3. Service Quality | SEQ1: Combine online and offline services (Online and Offline). |
| | SEQ2: Limited-time delivery service. |
| | SEQ3: Assist in fresh product management services. |
| 4. Playfulness | L1: Enjoy fresh food service. |
| | L2: Provide parent-child interactive entertainment activities. |
| 5. Epidemic Prevention | E1: Store personnel take temperature and monitor health. |
| | E2: Epidemic prevention education and training for store personnel. |
| | E3: Regular environment cleaning and disinfection. |
| 6. Service Convenience | C1: Combine with AI automated pick-up and packaging services. |
| | C2: One-package complete shopping service. |
| 7. Satisfaction | S1: Customer satisfaction with fresh products. |
| | S2: Customer's overall satisfaction with store service. |
| 8. Loyalty | LO1: Customers will continue to buy products in the store. |
| | LO2: Customers will continue to recommend the store to friends and family members. |
| 9. Purchase Intention | P1: Customers are willing to buy products in the store. |
| | P2: Customers spend more time and buy more products in the store. |

The research samples included 70 Fresh Hema stores in Shanghai that were located in different regions. Due to the different quarantine measures in each region, there were significant differences in consumer behavior among regional groups. Hence, a cluster sampling survey method was used to conduct an online survey questionnaire regarding the consumer behavior of Fresh Hema stores in Shanghai. The total number of samples was 950. A total of 832 questionnaires were collected and 806 were considered valid. This study used a 5-point Likert-type scale for measurement. According to the basic information, 532 (66%) respondents were female, 274 (34%) respondents were male, and 485 respondents (60.2%) were over 30 years old; 236 respondents (29.3%) graduated from university. There were 582 respondents (72.2%) who had been infected with COVID-19, and 708 respondents (87.8%) who had been quarantined. Among the 70 Fresh Hema clusters, the minimum sample size was 6, the maximum sample size was 28, and the average sample size was 11.51.

According to the PCA analysis, the variable results were reduced, and 21 explanatory variables were analyzed using descriptive statistics, as shown in Table 2. The findings indicate that the average number of explanatory variables for each questionnaire item ranged from 3.1 to 3.9. The standard deviation at the individual level was between 0.2 and 0.4. The standard deviation at the group level was between 0.3 and 0.5. The maximum value at the individual level was 5. The minimum value at the individual level was between 2 and 3. The maximum value at the group level was between 3.8 and 4.7. The maximum value at the group level ranged from 2.7 to 3.2.

**Table 2.** Descriptive Statistical Analysis of the Twenty-one Variables regarding Consumer Behavior of Fresh Hema Stores.

| Item | M | SD$_1$ | SD$_2$ | Maximum Value between Samples | Minimum Value between Samples | Maximum Value between Groups | Minimum Value between Groups |
|------|------|------|------|------|------|------|------|
| G1 | 3.68 | 0.36 | 0.42 | 5 | 2 | 4.12 | 3.15 |
| G2 | 3.58 | 0.28 | 0.35 | 5 | 3 | 3.88 | 3.02 |
| SQ1 | 3.62 | 0.23 | 0.31 | 5 | 3 | 3.92 | 2.82 |
| SQ2 | 3.31 | 0.32 | 0.36 | 5 | 3 | 4.14 | 2.58 |
| SQ3 | 3.59 | 0.35 | 0.41 | 5 | 2 | 3.96 | 2.85 |
| SEQ1 | 3.25 | 0.29 | 0.31 | 5 | 3 | 4.02 | 3.01 |
| SEQ2 | 3.52 | 0.32 | 0.36 | 5 | 2 | 4.28 | 2.89 |
| SEQ3 | 3.41 | 0.26 | 0.35 | 5 | 2 | 3.98 | 2.76 |
| L1 | 3.12 | 0.35 | 0.39 | 5 | 2 | 4.28 | 3.25 |
| L2 | 3.54 | 0.31 | 0.42 | 5 | 2 | 4.31 | 3.07 |
| E1 | 3.28 | 0.20 | 0.32 | 5 | 3 | 4.58 | 2.95 |
| E2 | 3.61 | 0.31 | 0.33 | 5 | 3 | 4.46 | 3.02 |
| E3 | 3.57 | 0.24 | 0.29 | 5 | 3 | 4.62 | 2.73 |
| C1 | 3.71 | 0.28 | 0.30 | 5 | 2 | 4.53 | 2.76 |
| C2 | 3.42 | 0.36 | 0.33 | 5 | 3 | 4.68 | 3.12 |
| S1 | 3.81 | 0.36 | 0.35 | 5 | 3 | 4.69 | 3.25 |
| S2 | 3.62 | 0.28 | 0.31 | 5 | 3 | 4.51 | 3.02 |
| LO1 | 3.82 | 0.35 | 0.31 | 5 | 2 | 4.38 | 3.07 |
| LO2 | 3.59 | 0.26 | 0.32 | 5 | 2 | 4.59 | 3.21 |
| P1 | 3.73 | 0.31 | 0.37 | 5 | 2 | 4.68 | 3.08 |
| P2 | 3.51 | 0.21 | 0.35 | 5 | 2 | 4.53 | 3.12 |

Note: M is the average of explanatory variables; SD$_1$ is the standard deviation of 806 consumers at the individual level. SD$_2$ is the standard deviation obtained by weighing the 70 stores at the overall level.

This study used multilevel structural equation modeling (MSEM) for empirical analysis. MSEM primarily consists of two models. One is the measurement model, which first establishes a dimension variant model for measurement indicators. It then defines dimension variant terms by testing measurement indicators with one or more dominant variables through confirmatory factor analysis. The second is the structural model, which explores the causal path relationship between exogenous and endogenous structural variables [43]. This study used the goodness-of-fit index (DFI) to determine the inherent structure of the two models and the goodness of fit of the overall structure model. In the goodness-of-fit index, the $X^2$ value of the overall goodness of fit divided by the degree of freedom (DF) to obtain the $X^2/DF$ value must be less than the goodness-of-fit standard value 3 [50]. GFI and AGFI must be greater than 0.8 [51], both CFI and NFI must be more than 0.9 [52], and RMSR must be smaller than 0.05 [53].

### 3.2.1. Measurement Model Analysis

The goodness-of-fit measurement in this study conforms to the DFI criteria. The measurement model goodness-of-fit results are summarized in Table 3.

**Table 3.** Measurement Model Goodness-of-fit Results.

| Goodness-of-Fit Index | Threshold Value | Measured Value | Result Determination |
|------|------|------|------|
| Ratio of $X^2$ to degrees of freedom ($X^2/DF$) | ≤3.00 | 2.22 | Acceptable |
| Goodness-of-fit index (GFI) | ≥0.80 | 0.95 | Acceptable |
| Adjusted Goodness-of-fit index (AGFI) | ≥0.80 | 0.92 | Acceptable |
| Normed fit index (NFI) | ≥0.90 | 0.96 | Acceptable |
| Comparative fit index (CFI) | ≥0.90 | 0.93 | Acceptable |
| Root Mean Square Residual (RMSR) | ≤0.05 | 0.036 | Acceptable |

The standardized factor loading of each measurement item in this study is greater than 0.5 and the absolute value t is greater than 1.96. This indicates that the questionnaire content can accurately reflect this measurement model. Therefore, the empirical results of this study have convergent validity. The results show that the dimensions' composite reliability (CR) values were all greater than 0.7, indicating a high degree of reliability among the dimension indexes and a high degree of correlation among the measurement indexes. Meanwhile, the average variance extracted (AVE) was also greater than 0.5, which indicates that each index variable can effectively reflect its dimension variable. Therefore, the measurement model in this study has good reliability, as shown in Table 4.

**Table 4.** Reliability Analysis.

| Dimension Variable | Standardized Factor Loading | Standard Error (SE) | t Value | Composite Reliability (CR) | Average Variance Extracted (AVE) |
|---|---|---|---|---|---|
| 1. Green Logistics | | | | 0.936 | 0.786 |
| G1 | 0.865 | 0.021 | 15.36 | | |
| G2 | 0.932 | 0.034 | 12.32 | | |
| 2. System quality | | | | 0.925 | 0.875 |
| SQ1 | 0.931 | 0.023 | 14.62 | | |
| SQ2 | 0.886 | 0.012 | 16.32 | | |
| SQ3 | 0.925 | 0.028 | 13.75 | | |
| 3. Service quality | | | | 0.956 | 0.902 |
| SEQ1 | 0.906 | 0.028 | 12.36 | | |
| SEQ2 | 0.871 | 0.035 | 15.21 | | |
| SEQ3 | 0.936 | 0.021 | 14.31 | | |
| 4. Playfulness | | | | 0.928 | 0.879 |
| L1 | 0.895 | 0.031 | 14.36 | | |
| L2 | 0.902 | 0.024 | 11.32 | | |
| 5. Epidemic prevention | | | | 0.938 | 0.901 |
| E1 | 0.962 | 0.016 | 15.32 | | |
| E2 | 0.926 | 0.027 | 12.68 | | |
| E3 | 0.906 | 0.031 | 13.71 | | |
| 6. Service convenience | | | | 0.918 | 0.906 |
| C1 | 0.964 | 0.028 | 14.38 | | |
| C2 | 0.922 | 0.021 | 11.62 | | |
| 7. Satisfaction | | | | 0.941 | 0.865 |
| S1 | 0.891 | 0.025 | 9.77 | | |
| S2 | 0.906 | 0.019 | 12.36 | | |
| 8. Loyalty | | | | 0.928 | 0.906 |
| LO1 | 0.937 | 0.031 | 11.28 | | |
| LO2 | 0.928 | 0.028 | 13.25 | | |
| 9. Purchase Intention | | | | 0.912 | 0.892 |
| P1 | 0.896 | 0.014 | 12.27 | | |
| P2 | 0.901 | 0.029 | 13.21 | | |

Construct reliability (CR) = (sum of standardized loading) $^2$/[(sum of standardized loading) $^2$ + (sum of indicator measurement error)]. Indicator measurement error can be calculated as $1 -$ (standardized loading) $^2$. Average variance extracted (AVE) = (sum of squared standardized loadings)/[(sum of squared standardized loadings) + (sum of indicator measurement error)].

### 3.2.2. Path Coefficient Analysis

According to the goodness-of-fit results of this study's structural model, the data presented in Table 5 shows that each goodness-of-fit index meets the threshold requirements, indicating that the structural model displays a good fit.

**Table 5.** Structural Model Goodness-of-fit Results.

| Goodness-of-Fit Index | Threshold Value | Measured Value | Result Determination |
|---|---|---|---|
| Ratio of $X^2$ to degrees of freedom ($X^2/DF$) | $\leq 3.00$ | 2.36 | Acceptable |
| Goodness-of-fit index (GFI) | $\geq 0.80$ | 0.93 | Acceptable |
| Adjusted goodness-of-fit index (AGFI) | $\geq 0.80$ | 0.91 | Acceptable |
| Normed fit index (NFI) | $\geq 0.90$ | 0.95 | Acceptable |
| Comparative fit index (CFI) | $\geq 0.90$ | 0.92 | Acceptable |
| Root mean square residual (RMSR) | $\leq 0.05$ | 0.028 | Acceptable |

The path estimation results of this study's empirical analysis are shown in Figure 2.

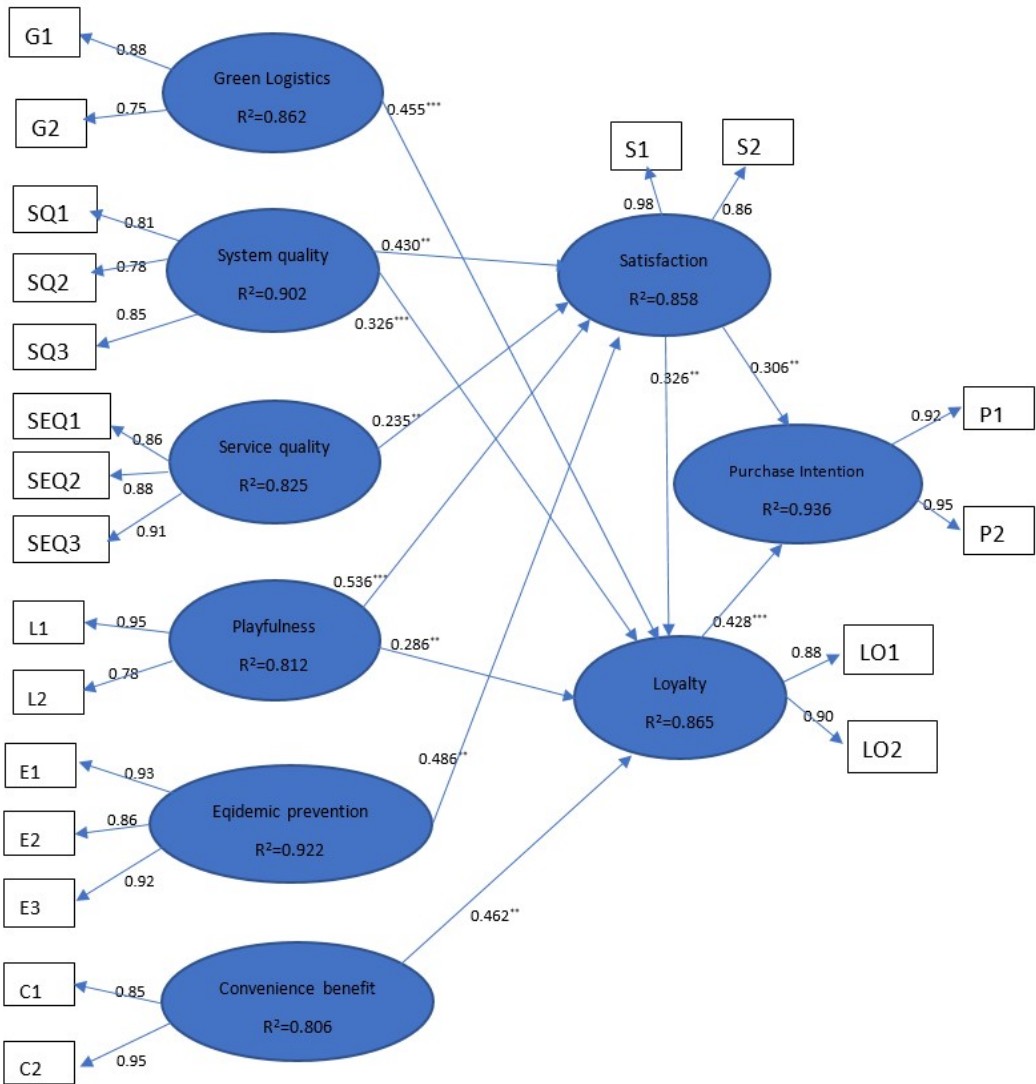

**Figure 2.** Estimate Results of the Path Coefficient of the Research Model. *** indicates $p < 0.01$; ** indicates $p < 0.05$.

As shown by the path coefficients and significance in Figure 2, all path coefficient estimates are consistent with the expected assumptions and are significant. This finding means that the 11 hypotheses proposed in this study are supported. According to each path coefficient, the impact of different dimensions on each other can be calculated. First, playfulness had the most significant impact on customer satisfaction. Its coefficient value was 0.536, indicating that customers attached importance to the stores' entertainment

and leisure value. The second was customer satisfaction with epidemic prevention. The coefficient value was 0.486, which indicates that customers attached great importance to epidemic prevention measures and safety maintenance of stores during the epidemic period. Third, convenience had a positive impact on loyalty. The coefficient value was 0.462. We estimated that due to the impact of the epidemic, stores combined the functions of the app and physical stores to provide one-package shopping services, from ordering and picking up goods to delivering goods to home, to avoid infection caused by human contact. Convenient shopping led to a positive impact on customer loyalty. Fourth, green logistics had a positive impact on loyalty. The coefficient value was 0.455, indicating that consumers attached importance to environmental protection, green energy, and carbon reduction, while affirmating fast logistics transportation.

Regarding the positive impact of the functional variables provided by the online fresh food shopping malls on satisfaction, the impact from the largest to the smallest was: playfulness, epidemic prevention, system service, and service quality. This indicates that the current consumer behavior has been transformed, and consumers have begun to attach importance to entertainment, epidemic prevention, and safety. The service quality to which traditional stores attached importance had minimal impact. Additionally, among dimension variables that positively impact customer loyalty, their order of impact was: convenience benefit, green logistics, system quality, and playfulness. This indicates that the convenience benefit had the most significant impact on loyalty. Consumers want to shop in the shortest time and at the lowest cost. Consumers have environmental awareness and pay attention to the green logistics concept of green energy and carbon reduction. Among the structural variables that positively impact purchase intention, loyalty had the most significant impact, indicating that stores should pay attention to improving customer loyalty to increase store revenue.

Based on the above analysis, as advances in digital technology and environmental changes lead to changes in personal consumption patterns, online fresh food shopping malls can grasp new trends and the key to market success by mastering consumer behavior. Kedah, Ismail, Ahasanul, and Ahmed [54] and Ali and Alfaki [55] make a similar point to show the relationship between playfulness and consumer satisfaction and loyalty. According to the empirical results obtained in this study, online fresh food shopping malls must consider consumption playfulness, epidemic prevention, safety maintenance, green energy, environmental protection logistics, etc., to effectively increase consumer satisfaction, loyalty, and purchase intention, and thus to increase store performance and revenue.

The empirical results found that the most important potential variables affecting consumer loyalty during the COVID-19 epidemic were convenience and green logistics services, and the relationships were positive, with impact coefficients of 0.462 and 0.455, respectively. This meant that, in order to avoid the risk of catching the virus in groups, consumers preferred free delivery logistics services and efficient shopping services that used AI technology to quickly pick up and deliver goods. In addition, the factors that have the greatest impact on consumer satisfaction are the playfulness effect and the epidemic prevention effect. The influence coefficients are 0.536 and 0.486, respectively, and both have positive relationships. T shows that consumers believe that, although it is inconvenient to go out shopping, they can choose to enjoy shopping through online shopping and online video interaction, taking into account epidemic prevention safety and shopping needs. In the MSEM path relationship diagram, it is found that there are two potential variables that affect customer purchase intention, namely loyalty and satisfaction, and loyalty has a greater effect on purchase intention, with influence coefficients of 0.482 and 0.306, respectively. From the above direct and indirect effects, it is found that because Fresh Hema mainly sells the daily necessities of fresh produce and the daily necessities of people's livelihoods, during the COVID-19 epidemic prevention period, consumers valued Fresh Hema's provision of free delivery and customer service under the AI technology the most, which is also the most important factor that affects the consumers' willingness to buy.

Under the new retail industry model, Fresh Hema integrates the online and offline sales models, and if it can provide customers with the most necessary online and offline customer service, it will increase the supply and inventory turnover rate of fresh agricultural products, which not only maintains the quality of agricultural products and avoids waste, but also enhances the one-stop, most efficient service of the agricultural products from the production end to the customer end, increasing the value of the overall agricultural products supply chain.

## 4. Conclusions and Recommendations

This study investigates the effects of green logistics, system quality, service quality, playfulness, safety and epidemic prevention, convenience benefits, customer satisfaction, and loyalty on the purchase intention of Alibaba's Fresh Hema in Mainland China. The hypotheses proposed in this study are all supported by the results of structural equation modeling. Among them, the variable of playfulness has the greatest positive effect on customer satisfaction, followed by the variable of epidemic prevention and safety, which indicates that consumer behavior has changed due to the increasing development of smart technology and the impact of the epidemic. In other words, from the point of view of playfulness, consumers tend to be attracted to interesting and joyful things. Therefore, it is recommended that online fresh food shopping malls adopt game-based marketing as one of the feasible strategies in the future. For example, online fresh food shopping malls can grasp the principle of "big prizes and lots of small prizes" by means of bonus point rewards, cash rebates for spending, friend referral gifts, achievement medals, or leaderboard competitions, etc., in order to make online shopping more fun, thereby increasing the degree of interaction between the mall and its customers and further enhancing customer satisfaction. In addition, as AI technology allows businesses to further integrate physical and online stores, providing a seamless online mall shopping experience for agricultural products is also a viable direction for the future. For example, the recommendation engine driven by AI technology can accurately pinpoint each consumer's preference for agricultural products and even eating habits, and further generate more personalized recommendation content when consumers browse the mall so that the appropriate products can be recommended to the most suitable consumers in the shortest possible time.

In addition, among the potential variables affecting customer loyalty, the ones with the greatest effects are the convenience benefit and green logistics, indicating that consumers place the greatest importance on the convenience benefit of shopping, as well as green awareness, in their busy lives. Therefore, we recommend that online fresh food malls strengthen the service function of customers in the front office to improve the convenience and safety of consumers using the mall. This can include providing detailed information, such as agricultural product pesticide inspections, quality certifications, product histories, and carbon footprint labels, so that consumers can view the information when shopping. In addition to compensating for consumers' concerns that agricultural products are not easy to standardize, this would enable consumers who care about green environmental issues to also obtain relevant information and buy products more securely. Meanwhile, petty farmers could also pass the characteristics of their farm products to consumers through this process. Regarding the back office of the online fresh food mall, the return and exchange mechanism, payment security, and logistics distribution should be adequately planned to reduce the unpredictability of consumers searching for agricultural products. This means that through the continual exchange between the three elements of people, stores (channels), and things (goods), the consumer experience is ultimately enhanced, thus improving consumer satisfaction and loyalty.

## 5. Limitations and Future Research

Only empirical analyses on Fresh Hema were conducted in this study. However, Fresh Hema has more than 300 stores in 27 cities across China. Due to sample limitations, it was not possible to analyze stores in more cities. Specifically, China has a vast territory, and

the consumption behavior in first- and second-tier cities may differ due to geographical factors. Therefore, this study only analyzed the available collected store big data, with limitations in the making of inferences about populations based on samples. Presently, there are various fresh product e-commerce companies in China, including traditional and new fresh food stores. New fresh food stores include home delivery, to-store + home delivery, pickup at the drop box, community group purchases, and other models. Different business models of fresh product e-commerce companies meet consumer needs at varying levels. We suggest that in addition to continuously exploring consumer loyalty and satisfaction, fresh product e-commerce companies in China should further explore a sustainable development economic model. Specifically, fresh product e-commerce is not widely used online. Thus, how to create branded and highly differentiated frozen food under the new agricultural value chain to achieve a reasonable gross profit margin is a topic worthy of further research.

**Author Contributions:** Methodology, C.-C.Y.; Data curation, C.-C.Y.; Writing—original draft, C.-S.S.; Writing—review & editing, C.-S.L. All authors have read and agreed to the published version of the manuscript.

**Funding:** This research received no external funding.

**Institutional Review Board Statement:** All subjects in the study were anonymously labeled and agreed to participate in the survey.

**Data Availability Statement:** No data support.

**Conflicts of Interest:** The authors declare no conflict of interest.

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
