# Peer review of "The Impact of Consumer Loyalty and Customer Satisfaction in the New Agricultural Value Chain"

_agriculture, doi:10.3390/agriculture13091803_

Round 1
Reviewer 1 Report
Dear Authors,
Thanks for your stimulating article.
I do think that your analysis is based on a sound statistical approach. I suggest a minor point that can add some extra value to your article. That is to introduce additional details about the retail model you analyze in this study.
Also, I would like to read about the sample description in the methodology section of the paper.
Furthermore, you told in your introduction that: "This paper also explores shopping malls' and petty farmers' production and marketing strategies under the latest retail trends." Consequently, I want to read more robust and profound discussion and conclusion sections.
Some minor expressions and phrasal verb usage could be improved using a native proofreading process. However, this is not at all a must.
Author Response
Reviewers’ Comments and Suggestions for Author (Round 1)
5 September 2023
Dear Reviewer,
Thank you for the constructive suggestions and comments on our manuscript(ID: agriculture-2574699). The suggestions and comments are helpful for improving the manuscript. We are submitting the revised version of the manuscript with our responses to the suggestions and comments by the reviewer. Many thanks for your guidance.
Our responses to each suggestion and comment are as follows, and they are presented in blue texts with a grey background color in the revised manuscript.
Dear Authors,
Thanks for your stimulating article.
I do think that your analysis is based on a sound statistical approach. I suggest a minor point that can add some extra value to your article. That is to introduce additional details about the retail model you analyze in this study.
Also, I would like to read about the sample description in the methodology section of the paper.
Furthermore, you told in your introduction that: "This paper also explores shopping malls' and petty farmers' production and marketing strategies under the latest retail trends." Consequently, I want to read more robust and profound discussion and conclusion sections.
Response:
Thank you very much for your comments and suggestion. The modifications are as follows:
1.Here are additional details about the retail model you analyze in this study.
Compared with other department store products, fresh products have a low degree of standardization, a high rate of wear and tear, and a high degree of difficulty in quality control. In addition, the preservation conditions and transportation and storage requirements for fresh products are relatively high. Therefore, Fresh Hema emphasizes a low-cost cold chain home delivery model and builds an integrated online and offline supermarket, which means that online sales are the mainstay and offline sales are supplementary. In other words, Fresh Hema uses the offline store as the basis and the warehouse, sorting and distribution center of the online Fresh Hema APP, and satisfies the needs of consumers for fresh food purchasing, dining and leisure within the surrounding 3km area through the complete integration of online and offline business. Fresh Hema's advertising and marketing efforts are based on consumer behavioral data, attracting more consumers and creating a virtuous consumer cycle. Since China's domestic market demand is strong, examining how Fresh Hema can maintain good operations among fresh product e-commerce companies amid fierce competition is a research topic that should be further explored. iResearch (2021) indicated that Fresh Hema is a pioneer in the new retail format of "supermarket + catering." Fresh Hema is a brand owned by Alibaba, which provides online and online integrated services through a business model that combines offline stores using an online app. Offline stores and warehouses play a pre-warehouse function. Meanwhile, Fresh Hema combines supermarkets and catering to strengthen the capacity to meet the real-time needs for fresh products.
- Here are the sample description in the methodology section of the paper.
The research samples included 70 Fresh Hema stores in Shanghai that were located in different regions. Due to the different quarantine measures in each region, there were significant differences in consumer behavior among regional groups. Hence, a cluster sampling survey method was used to conduct an online survey questionnaire regarding consumer behavior of Fresh Hema stores in Shanghai. The total number of samples was 950. A total of 832 questionnaires were collected and 806 were considered valid. This study used a 5-point Likert-type scale for measurement. According to the basic information, 532 (66%) respondents were female, 274 (34%) respondents were male, and 485 respondents (60.2%) were over 30 years old; 236 respondents (29.3%) graduated from university. There were 582 respondents (72.2%) who had been infected with COVID-19, and 708 respondents (87.8%) who had been quarantined. Among the 70 Fresh Hema clusters, the minimum sample size was six, the maximum sample size was 28, and the average sample size was 11.51.
According to the PCA analysis, the variable results were reduced, and 21 explanatory variables were analyzed using descriptive statistics, as shown in Table 6.1. The findings indicate that the average number of explanatory variables for each questionnaire item ranged from 3.1 to 3.9. The standard deviation of the individual level was between 0.2 and 0.4. The standard deviation of the group level was between 0.3 and 0.5. The maximum value at the individual level was 5. The minimum value at the individual level was between 2 and 3. The maximum value at the group level was between 3.8 and 4.7. The maximum value at the group level ranged from 2.7 to 3.2.
Table 1: Descriptive Statistical Analysis of the twenty one Variables regarding Consumer Behavior of Fresh Hema Stores
|
Item |
M |
SD1 |
SD2 |
Maximum value between samples |
Minimum value between samples |
Maximum value between groups |
Minimum value between groups |
|
G1 |
3.68 |
0.36 |
0.42 |
5 |
2 |
4.12 |
3.15 |
|
G2 |
3.58 |
0.28 |
0.35 |
5 |
3 |
3.88 |
3.02 |
|
SQ1 |
3.62 |
0.23 |
0.31 |
5 |
3 |
3.92 |
2.82 |
|
SQ2 |
3.31 |
0.32 |
0.36 |
5 |
3 |
4.14 |
2.58 |
|
SQ3 |
3.59 |
0.35 |
0.41 |
5 |
2 |
3.96 |
2.85 |
|
SEQ1 |
3.25 |
0.29 |
0.31 |
5 |
3 |
4.02 |
3.01 |
|
SEQ2 |
3.52 |
0.32 |
0.36 |
5 |
2 |
4.28 |
2.89 |
|
SEQ3 |
3.41 |
0.26 |
0.35 |
5 |
2 |
3.98 |
2.76 |
|
L1 |
3.12 |
0.35 |
0.39 |
5 |
2 |
4.28 |
3.25 |
|
L2 |
3.54 |
0.31 |
0.42 |
5 |
2 |
4.31 |
3.07 |
|
E1 |
3.28 |
0.20 |
0.32 |
5 |
3 |
4.58 |
2.95 |
|
E2 |
3.61 |
0.31 |
0.33 |
5 |
3 |
4.46 |
3.02 |
|
E3 |
3.57 |
0.24 |
0.29 |
5 |
3 |
4.62 |
2.73 |
|
C1 |
3.71 |
0.28 |
0.30 |
5 |
2 |
4.53 |
2.76 |
|
C2 |
3.42 |
0.36 |
0.33 |
5 |
3 |
4.68 |
3.12 |
|
S1 |
3.81 |
0.36 |
0.35 |
5 |
3 |
4.69 |
3.25 |
|
S2 |
3.62 |
0.28 |
0.31 |
5 |
3 |
4.51 |
3.02 |
|
LO1 |
3.82 |
0.35 |
0.31 |
5 |
2 |
4.38 |
3.07 |
|
LO2 |
3.59 |
0.26 |
0.32 |
5 |
2 |
4.59 |
3.21 |
|
P1 |
3.73 |
0.31 |
0.37 |
5 |
2 |
4.68 |
3.08 |
|
P2 |
3.51 |
0.21 |
0.35 |
5 |
2 |
4.53 |
3.12 |
Note: M is the average of explanatory variables; SD1 is the standard deviation of 806 consumers at the individual level. SD2 is the standard deviation obtained by weighing the 70 stores at the overall level.
3.Here are robust and profound discussion and conclusion sections.
This phenomenon has had an impact on the agricultural value chain. This study found that Fresh Hema stores had the same prices online and offline. However, fresh product from e-commerce platforms could not provide consumers with a consumption experience comparable to traditional fresh products. Therefore, information regarding product origin, packaging, specifications, and transportation can be disclosed on e-commerce platforms with a page design that attracts consumers' attention. In addition to attracting consumers' attention, page design can make consumers more willing to pay for high-quality fresh food through a comprehensive platform experience and direct audio-visual interaction. This has also created a trend of actively optimizing playfulness through e-commerce platforms selling fresh products. In the post-epidemic era, consumers have got over the fear over the pandemi, and epidemic prevention and safety have been stable. According to the psychology of subsequent compensation, the public will eventually return to the rational new normal; the mastery of price sensitivity of fresh product e-commerce platforms is also a trend that must be addressed in the future.
Here is the impact on local petty farmers. At the beginning of each year, Fresh Hema conducts a unified inventory and plan for geotagged agricultural products, conducts research on local petty farmers or agricultural products in advance, analyzes their capacity, place of production, market demand, etc., and then the person in charge of the procurement business connects with them one by one, and conducts marketing and promotion of different brands. Most of Fresh Hema's agricultural products on the shelves are protected by geotagging, and the consumers have a high degree of recognition of this mark, which has also led to the increase in income of the local petty farmers.

Reviewer 2 Report
I like this issue, but this paper is too simple to study the impact of consumer loyalty and customer satisfaction in new agricultural value chain. Here are some suggestions of your research:
- Introduction:
- Begin by providing a clear context for your study. Explain the importance of understanding consumer behavior in the changing landscape of fresh agricultural product purchasing, influenced by AI and environmental awareness.
- Research Objectives:
- Explicitly state your research objectives. Clarify that your study aims to investigate the impact of consumer loyalty and satisfaction in the new agricultural value chain, specifically focusing on Fresh Hema.
- Literature Review:
- Provide a comprehensive review of existing literature on consumer behavior in the context of agricultural value chains, new retail models, and the role of AI and environmental awareness. Highlight the research gaps that your study intends to address.
- Rationale for Methodology:
- Explain your rationale for employing Principal Component Analysis (PCA) and Multilevel Structural Equation Modeling (MSEM). Justify why these methods are well-suited for analyzing consumer behavior and relationships between variables.
- Methodology Section:
- Elaborate on the steps taken in conducting PCA. Detail how essential factors of online consumer behavior were identified and selected. Mention any criteria used to retain principal components.
- Describe the process of applying MSEM. Mention how you incorporated different levels of the new retail model. Explain why MSEM is appropriate for studying relationships among variables.
- Data Collection and Sample:
- Specify the source of your data, which is online questionnaires collected from Fresh Hema consumers in 2022. Briefly explain why this dataset is relevant to your study.
- Provide details about your sample size, sampling method, and any demographic information collected from respondents.
- Results and Interpretation:
- Present the main findings of your analysis. Clearly state the impacts of playfulness, epidemic prevention, convenience benefits, and green logistics on customer satisfaction and loyalty.
- Offer explanations for the observed results based on both theoretical reasoning and the context of Fresh Hema.
- Discussion and Implications:
- Interpret the implications of your findings for the agricultural value chain and the new retail model. Discuss how your results align with the broader trends in consumer behavior and the changing retail landscape.
- Speculate on why convenience benefits and green logistics have a more significant impact on customer loyalty compared to other factors.
- Comparison with Prior Research:
- Compare your findings with relevant previous studies. Highlight any consistencies or disparities, and discuss potential reasons for these differences.
- Limitations and Future Research:
- Acknowledge the limitations of your study, such as potential biases in self-reported data or limitations specific to the chosen methods. Suggest avenues for future research that could overcome these limitations.
- Conclusion:
- Summarize the main takeaways from your study. Reiterate the significance of understanding consumer loyalty and satisfaction in the evolving agricultural value chain.
- References:
- Ensure that you provide a comprehensive list of references to support your study. Make sure to include the source from where you obtained the online questionnaire.
For the improvement of the methodology, the suggestions are:
- Elaborate on the Method Selection:
- Start by providing a rationale for why you chose PCA and MSEM as your chosen methods. Explain how each method addresses specific aspects of your research objectives. Highlight the advantages of using these methods in studying complex consumer behavior phenomena.
- PCA Procedure:
- Detail the step-by-step process of PCA. Discuss data preprocessing steps (such as data cleaning, scaling, and normalization) and the specific PCA algorithm you used. Highlight any considerations you took regarding the number of components to retain or the explained variance threshold.
- Interpretation of PCA Components:
- After applying PCA, provide a detailed interpretation of the identified principal components. Explain how these components capture the essential factors of online consumer behavior. Relate these components back to the theoretical constructs in your research.
- MSEM Model Specification:
- Describe the structural equation model you constructed in more detail. Elaborate on the theoretical underpinnings of the variables included in the model. Explain why certain variables were chosen and how they relate to the concepts of customer loyalty and satisfaction.
- MSEM Estimation and Fit Measures:
- Provide a comprehensive overview of the estimation process in MSEM. Explain the statistical software you used and any relevant syntax or commands. Discuss how you assessed the model fit using various fit indices (e.g., CFI, RMSEA) and provide cutoff criteria for determining acceptable fit.
- Handling of Missing Data and Assumptions:
- Address the issue of missing data if applicable. Explain your approach for handling missing values and its potential impact on the results. Discuss any assumptions of MSEM (e.g., normality, linearity) and how you assessed their validity.
- Results Interpretation:
- Expand upon the interpretation of your MSEM results. Discuss the magnitude, direction, and significance of the relationships between the selected variables. Relate these findings back to your research hypotheses and the theoretical framework.
- Model Comparison and Sensitivity Analysis:
- Consider conducting sensitivity analyses or model comparisons. This could involve comparing different models with varying sets of variables or testing alternative specifications. Discuss how these additional analyses support the robustness of your findings.
- Discussion of Implications:
- Extend the discussion of your findings to include the practical implications of the results. How can the identified factors impact consumer behavior and the performance of the new agricultural value chain? Discuss how businesses or policymakers could leverage these insights.
- Limitations and Future Research:
- Clearly outline the limitations of your methodology. Discuss potential sources of bias, limitations of the chosen methods, and constraints in generalizing the results. Suggest avenues for future research that could build upon or address these limitations.
- Conclusion of Methodology:
- Summarize the strength and appropriateness of using PCA and MSEM for your study. Reiterate how the combined use of these methods allowed you to comprehensively investigate the impact of consumer loyalty and satisfaction in the new agricultural value chain.
The suggestions are:
- Language and Clarity: Ensure that the language used is clear, precise, and easily understandable. Avoid unnecessary jargon and technical terms that might confuse readers.
- Tables and Figures: If relevant, consider including tables or figures to present the data in a visual and organized manner, making it easier for readers to comprehend the findings.
- Formatting and Citations: Follow the formatting guidelines of the target journal and ensure that all sources are properly cited using the appropriate citation style.
- Grammar and Proofreading: Thoroughly proofread the article to correct any grammatical errors or typos. Seek feedback from colleagues or peers to ensure the clarity and quality of the writing.
Author Response
Reviewers’ Comments and Suggestions for Author (Round 1)
5 September 2023
Dear Reviewer,
Thank you for the constructive suggestions and comments on our manuscript(ID: agriculture-2574699). The suggestions and comments are helpful for improving the manuscript. We are submitting the revised version of the manuscript with our responses to the suggestions and comments by the reviewer. Many thanks for your guidance.
Our responses to each suggestion and comment are as follows, and they are presented in blue texts with a grey background color in the revised manuscript.
Response:
Thank you very much for your comments and suggestion. The modifications are as follows:
- Introduction:
- Begin by providing a clear context for your study. Explain the importance of understanding consumer behavior in the changing landscape of fresh agricultural product purchasing, influenced by AI and environmental awareness.
Regarding fresh agricultural product purchasing, there are a growing number of choices for consumers. Even top e-commerce brands have little difference in the sources of their products. Therefore, how to trigger the purchase intention of consumers who buy fresh agricultural products and enhance their loyalty after entering an e-commerce website can affect marketing and advertising efficiency, which may lead to differing costs. The development of AI enables fresh agricultural products e-commerce companies to perform data management and smart tracking while collecting and cross-analyzing the view history of each visitor on their website. For example, during COVID-19, consumers' shopping methods changed. AI data analysis indicates that the turnover of frozen agricultural food packages increased significantly. This has encouraged e-commerce companies to build automated product recommendation mechanisms to understand and promote consumer loyalty. Therefore, examining consumer behavior regarding fresh agricultural products has significant benefits for e-commerce, allowing AI to determine the best product recommendation based on consumer behavior. Therefore, it is important to understand the consumer behavior of fresh agricultural products.
- Research Objectives:
Explicitly state your research objectives. Clarify that your study aims to investigate the impact of consumer loyalty and satisfaction in the new agricultural value chain, specifically focusing on Fresh Hema.
Line 90-109
According to the Report on China's Fresh Food E-commerce Market Data in H1 2023, the scale of fresh food e-commerce transactions in China is expected to reach RMB 642.76 billion in 2023. Additionally, in H1 2023, the fresh food e-commerce transactions of urban residents in China accounted for 27.6% of urban residents' food consumption expenditure. The major e-commerce companies in Shanghai include Dingdong Maicai, Jingdong Daojia, RT-Mart Youxian, Duoduo Maicai, Fruitday, and Hema Taiwan Trade Center in Shanghai [9]. Fresh Hema currently coexists with competitors using multiple business models. Fresh quality rather than price is the key factor in first- and second-tier cities in China. Thus, how to provide better shopping experience for in-store consumption or online shopping through big data while achieving instant delivery to improve performance efficiency is a top priority for the new agricultural value chain. These are what consumers are truly concerned about. As such, the purpose of this study was to explore the correlation between Fresh Hema's value-creating strategy and consumer satisfaction. On the other hand, the dividend of Internet traffic is gradually disappearing, and the cost of website traffic is increasing. Fresh Hema creates the value of its products and services through Hema physical stores and catering services, enabling consumers to place orders through online apps and consume offline. However, with many alternatives to fresh agricultural products and if branding is low, once a consumer is dissatisfied, such as receiving stale products, they may turn to other e-commerce merchants for consumption. Therefore, examining consumer loyalty of Fresh Hema products was also a focus of this study.
- Literature Review:
Provide a comprehensive review of existing literature on consumer behavior in the context of agricultural value chains, new retail models, and the role of AI and environmental awareness. Highlight the research gaps that your study intends to address.
Line 118-135
Yan, Chen, Cai, and Guan [10] reported that the fresh agricultural product supply chain's circulation efficiency is significantly influenced by the purchasing power of end consumers. Consumers are an essential factor in determining agricultural value chains. Mowat and Collins [11] found that supply chains in new and emerging agricultural industries typically lack the ability to link product quality with consumer behaviour.
Agricultural value chains have also changed over time. Dong [12] found that agricultural value chains are going through a paradigm shift from efficiency-driven to resilience-focused eco-friendly agriculture. Other factors affecting agriculture value chains include AI. Ganeshkumar, Jena, Sivakumar, and Nambirajan [13] indicated that with AI adoption in agriculture, value chains can increase agricultural income, enhance competitiveness, and reduce cost. Among the agriculture value chains stages, AI research related to agricultural processing and consumer sector is limited compared to input, production, and quality testing. Based on the above relevant literature, it is evident that consumer behavior plays an essential role in agriculture value chains. The present study focuses on the increasing trend of consumers' demand for fresh agricultural products after COVID-19, addressing the gap in previous studies that primarily focused on general agricultural products [14-16]. This study also examines which parties are most suited to facilitate value chain upgrading.
- Rationale for Methodology:
Explain your rationale for employing Principal Component Analysis (PCA) and Multilevel Structural Equation Modeling (MSEM). Justify why these methods are well-suited for analyzing consumer behavior and relationships between variables.
In the research on consumer behavior, the estimation of variables is primarily reduced through Principal Component Analysis (PCA). This process can objectively extract important representative explanatory variables and resolve the possibility of autocorrelation among explanatory variables that could lead to statistical fallacy. When researchers conduct factor analysis with specific restrictions on the structure of factors, it is called confirmatory factor analysis (CFA), which belongs to the SEM model (Chiou, 2003; Yu, 2006). If the essential structure of latent variables estimated by the traditional SEM model is questioned, the best method is to use CFA to test the difference between the factor model at the individual level and at the overall level. Based on statistical inference, the MSEM estimation method conforms to the conditions that in the multi-level theory, the latent variables are unbiased. As such, the data can be further analyzed without restriction on the extraction of latent variables and the structural analysis of consumer behavior factors at multiple levels (Hema mall stores or different epidemic areas).
This study explored the impact of COVID-19 on consumer behavior of Fresh Hema. An effective way to avoid the spread of COVID-19 was to conduct quarantine measures in each epidemic area. Therefore, this study examined the distribution of 70 Fresh Hema stores in various regions in Shanghai in China and conducted research based on 70 consumer clusters. Due to the different degree of COVID-19 in each region, consumers displayed various motives and intentions to purchase in Fresh Hema stores. Based on the study's sample data, the results indicated large differences between clusters (large variance) and small differences within groups (small variance), which met the cluster sampling criteria.
The main difference between single-level and multi-level data analysis is that multi-level data structure divides low-level individuals into different clusters according to a certain grouping variable area. Then, it analyzes the inter-cluster and intra-cluster variance of the analyzed variable (Y), respectively. Both SEM and MSEM are based on the covariant structure of the variables; operations are handled in terms of discrete fractions of variables (Y- ), i.e., centering or deviated scores, denoted by a lowercase y (Cronbach & Webb, 1979).
For a variable after centering, the origin of the measure is translated to the mean position. Therefore, the intercept obtained in the integrated processing in SEM mode is equivalent to the average value. Additionally, since the inter-cluster and intra-cluster centering scores (deviation fractions) are orthogonal fractions, two independent and additive covariant matrices can be derived — the intra-group covariant matrix (SW) and the inter-cluster covariant matrix (SB):
The SW and SB obtained from the sample observations are additive. The sum of the intra-cluster variance and the sum of the inter-cluster variance obtained from the variables in the vector are unbiased estimates of the intra-cluster variance () and inter-cluster variance (), respectively. Where, w and b with lower case subscripts represent the variance scalar accumulated by each variable since the expansion of the matrix. Therefore, the ratio of the inter-cluster variance of each variable to the total variance can be calculated as the ICC (Intra-Class Coefficient).
is the proportion of the difference between the clusters in the variance of the measured variable, i.e., the inter-cluster effect. When is very small, this means that the differences between groups are not obvious, the influence of multi-level structure can be ignored, and the data can be addressed using traditional SEM methods. When the is very large, this means that the differences between clusters are too large to be ignored, and MSEM technology must be used to address them (Roberts, 2002).
Based on the data of 70 Fresh Hema stores in Shanghai, the measured by cluster was 0.68. Roberts (2002) suggested that if the value was greater than 0.5, MSEM technology should be used to analyze the data. Therefore, MSEM was used in this study to measure the consumer behavior analysis of 70 Fresh Hema stores in Shanghai.
Although this study used principal component analysis (PCA) to extract important variables objectively and define the sum of explanatory variables of latent variables, we also avoided statistical analysis fallacies since explanatory variables may have autocorrelation. In this process, the influence of measurement error may also be ignored, which will lead to the reduction of statistical test power (Bollen, 1989). Therefore, we also used CFA to test the underlying factor structure of contextual variables at the overall level to determine the applicability of the structural model and overcome the limitations of traditional MLM.
References
Yu, M. N. (2006). “Latent Variable Models: The Application of SIMPLIS”. Taipei City: Higher Education Press.
Chiou, H. J. (2003). “Principles and practice of structural equation modeling with LISREL”. Taipei City: Yeh Yeh Book Gallery Ltd.
- Methodology Section:
- Elaborate on the steps taken in conducting PCA. Detail how essential factors of online consumer behavior were identified and selected. Mention any criteria used to retain principal components.
This study conducted Principal Component Analysis (PCA) according to the 35 questionnaire study questions. According to the PCA results, we retained 21 explanatory variables with factor loading greater than 0.5. Hair et al. suggested that if (PCA) factor loading is below 0.5, the explanatory variable should be deleted (Hair et al., 2006). Based on the above, this study defined nine dimension variable names by retaining the characteristics of the explanatory variables. Table 5.1 shows that the cumulative explanatory variance of the nine dimension variables was 88.65%. This indicates that the dimension variables were highly representative and explanatory. Furthermore, Cronbach's alpha values of the dimension variables were all higher than 0.8, indicating high reliability.
Table 5.1 Principal Component Analysis (PCA) of the Explanatory Variables
|
Questionnaire Variable |
Factor Loading |
Eigenvalue |
Cumulative Explanatory Variance (%) |
Cronbach’s alpha |
|
Dimension Variable 1: Green Logistics |
|
4.26 |
18.5 |
0.92 |
|
G1 Use environmentally friendly packaging |
0.865 |
|
|
|
|
G2 Use environmentally friendly shopping bags |
0.932 |
|
|
|
|
Dimension Variable 2: System quality |
|
3.85 |
33.86 |
0.89 |
|
SQ1 Combine network video systems |
0.931 |
|
|
|
|
SQ2 Provide product information online |
0.886 |
|
|
|
|
SQ3 APP mobile payment service |
0.925 |
|
|
|
|
Dimension Variable 3: Service Quality |
|
3.62 |
46.72 |
0.90 |
|
SEQ1 Combine online and offline services |
0.906 |
|
|
|
|
SEQ2 Limited time delivery service. |
0.871 |
|
|
|
|
SEQ3 Assist in fresh product management services. |
0.936 |
|
|
|
|
Dimension Variable 4: Playfulness |
|
3.72 |
57.68 |
0.88 |
|
L1 Enjoy fresh food service. |
0.895 |
|
|
|
|
L2 Provide parent-child interactive entertainment activities. |
0.902 |
|
|
|
|
Dimension Variable 5: Epidemic Prevention |
|
3.81 |
66.42 |
0.86 |
|
E1 Store personnel take temperature and monitor health. |
0.962 |
|
|
|
|
E2 Epidemic prevention education and training for store personnel. |
0.926 |
|
|
|
|
E3 Regular environment cleaning and disinfection. |
0.906 |
|
|
|
|
Dimension Variable 6: Service Convenience |
|
3.51 |
74.25 |
0.87 |
|
C1 Combine with AI automated pickup and packaging services. |
0.964 |
|
|
|
|
C2 One-package complete shopping service. |
0.922 |
|
|
|
|
Dimension Variable 7: Satisfaction |
|
3.36 |
80.98 |
0.85 |
|
S1 Customer satisfaction with fresh products. |
0.891 |
|
|
|
|
S2 Customer's overall satisfaction with store service. |
0.906 |
|
|
|
|
Dimension Variable 8: Loyalty |
|
3.51 |
85.27 |
0.83 |
|
LO1 Customers will continue to buy products in the store. |
0.937 |
|
|
|
|
LO2 Customers will continue to recommend friends and family members to the store. |
0.928 |
|
|
|
|
Dimension Variable 9: Purchase Intention |
|
3.32 |
88.65 |
0.84 |
|
P1 Customers are willing to buy products in the store. |
0.896 |
|
|
|
|
P2 Customers spend more time and buy more products in the store. |
0.901 |
|
|
|
|
Total cumulative explanatory variance (%) |
|
|
88.65 |
|
References
Hair, J. F., Black, W. C., Babin, B. J., Anderson, R. E., & Tatham, R. L. (2006). Multivariate data analysis (6th ed.). Prentice-Hall.
- Describe the process of applying MSEM. Mention how you incorporated different levels of the new retail model. Explain why MSEM is appropriate for studying relationships among variables.
Line 311-327
Due to the different locations of Fresh Hema stores and the extent of COVID-19, a combination of online and physical customer services under the new retail model also affects consumers' purchase motivation and behavior. It also indirectly affects consumers' satisfaction, loyalty, and purchase intention. During the epidemic, quarantine measures were implemented in various regions, resulting in multilevel and clustered consumer behavior data. If the traditional SEM model is used, the statistical observations will be invalid due to special dependence or duality, which will cause the violation of the hypothesis of sample independence and statistical tests. Therefore, this study used MSEM to analyze the influence of consumers' purchase motivation behavior on consumer satisfaction, loyalty, and purchase intention under the new retail model.
In this study, 70 Fresh Hema stores in Shanghai were taken as the basic clusters and Multilevel Structural Equation Modeling (MSEM) was used to solve the above statistical error problem [45-49]. Furthermore, the consumption behavior at the individual level, as well as the impact of latent variables on the overall level of latent dependent variables were correctly evaluated. This was performed to measure the customer service and functions that best meet the needs of consumers to effectively improve the value supply chain of fresh agricultural products.
References
Bentler, P. M., & Liang, J. (2003). Two-level mean and covariance structure: Maximum Likelihood via an EM algorithm. In S. P. Reise & N. Duan (Eds.), Multilevel modeling: Methodological advances, issues, and applications (pp. 53-70). Mahwah, NJ: Lawrence Erlbaum Associates.
Heck, R. H., & Thomas, S. L. (2000). An introduction to multilevel modeling techniques. Mahwah, NJ: Lawrence Erlbaum Associates.
Muthén, B. O. (1994). Multilevel covariance structure analysis. Sociological Methods and Research, 22, 376-398.
Hox, J. J. (2002). Multilevel analysis: Techniques and applications. Mahwah, NJ: Lawrence Erlbaum Associates.
Kaplan, D., & Elliott, P. R. (1997). A didactic example of multilevel structural equation modeling applicable to the study of organizations. Structural Equation Modeling, 4, 1-24.
- Data Collection and Sample:
Specify the source of your data, which is online questionnaires collected from Fresh Hema consumers in 2022. Briefly explain why this dataset is relevant to your study.
The data used in this study were obtained from the actual survey questionnaire of 70 Fresh Hema stores in Shanghai, China, from 2022. The purpose of this study was to explore the impact of COVID-19 on consumer behavior under the new retail model (O2O). According to the PCA analysis results, we redefined nine latent variables according to the meaning of the variables in the consumer behavior questionnaire (see Table 5.1). Additionally, we used the MSEN model to analyze the causal path relationships among the nine latent variables. We also analyzed the impact of latent independent variables of customer service provided by Fresh Hema on consumers' satisfaction, loyalty, and purchase intention (latent dependent variables) under the new retail model.
- Provide details about your sample size, sampling method, and any demographic information collected from respondents.
The research samples included 70 Fresh Hema stores in Shanghai that were located in different regions. Due to the different quarantine measures in each region, there were significant differences in consumer behavior among regional groups. Hence, a cluster sampling survey method was used to conduct an online survey questionnaire regarding consumer behavior of Fresh Hema stores in Shanghai. The total number of samples was 950. A total of 832 questionnaires were collected and 806 were considered valid. This study used a 5-point Likert-type scale for measurement. According to the basic information, 532 (66%) respondents were female, 274 (34%) respondents were male, and 485 respondents (60.2%) were over 30 years old; 236 respondents (29.3%) graduated from university. There were 582 respondents (72.2%) who had been infected with COVID-19, and 708 respondents (87.8%) who had been quarantined. Among the 70 Fresh Hema clusters, the minimum sample size was six, the maximum sample size was 28, and the average sample size was 11.51.
According to the PCA analysis, the variable results were reduced, and 21 explanatory variables were analyzed using descriptive statistics, as shown in Table 6.1. The findings indicate that the average number of explanatory variables for each questionnaire item ranged from 3.1 to 3.9. The standard deviation of the individual level was between 0.2 and 0.4. The standard deviation of the group level was between 0.3 and 0.5. The maximum value at the individual level was 5. The minimum value at the individual level was between 2 and 3. The maximum value at the group level was between 3.8 and 4.7. The maximum value at the group level ranged from 2.7 to 3.2.
Table 6.1: Descriptive Statistical Analysis of the twenty one Variables regarding Consumer Behavior of Fresh Hema Stores
|
Item |
M |
SD1 |
SD2 |
Maximum value between samples |
Minimum value between samples |
Maximum value between groups |
Minimum value between groups |
|
G1 |
3.68 |
0.36 |
0.42 |
5 |
2 |
4.12 |
3.15 |
|
G2 |
3.58 |
0.28 |
0.35 |
5 |
3 |
3.88 |
3.02 |
|
SQ1 |
3.62 |
0.23 |
0.31 |
5 |
3 |
3.92 |
2.82 |
|
SQ2 |
3.31 |
0.32 |
0.36 |
5 |
3 |
4.14 |
2.58 |
|
SQ3 |
3.59 |
0.35 |
0.41 |
5 |
2 |
3.96 |
2.85 |
|
SEQ1 |
3.25 |
0.29 |
0.31 |
5 |
3 |
4.02 |
3.01 |
|
SEQ2 |
3.52 |
0.32 |
0.36 |
5 |
2 |
4.28 |
2.89 |
|
SEQ3 |
3.41 |
0.26 |
0.35 |
5 |
2 |
3.98 |
2.76 |
|
L1 |
3.12 |
0.35 |
0.39 |
5 |
2 |
4.28 |
3.25 |
|
L2 |
3.54 |
0.31 |
0.42 |
5 |
2 |
4.31 |
3.07 |
|
E1 |
3.28 |
0.20 |
0.32 |
5 |
3 |
4.58 |
2.95 |
|
E2 |
3.61 |
0.31 |
0.33 |
5 |
3 |
4.46 |
3.02 |
|
E3 |
3.57 |
0.24 |
0.29 |
5 |
3 |
4.62 |
2.73 |
|
C1 |
3.71 |
0.28 |
0.30 |
5 |
2 |
4.53 |
2.76 |
|
C2 |
3.42 |
0.36 |
0.33 |
5 |
3 |
4.68 |
3.12 |
|
S1 |
3.81 |
0.36 |
0.35 |
5 |
3 |
4.69 |
3.25 |
|
S2 |
3.62 |
0.28 |
0.31 |
5 |
3 |
4.51 |
3.02 |
|
LO1 |
3.82 |
0.35 |
0.31 |
5 |
2 |
4.38 |
3.07 |
|
LO2 |
3.59 |
0.26 |
0.32 |
5 |
2 |
4.59 |
3.21 |
|
P1 |
3.73 |
0.31 |
0.37 |
5 |
2 |
4.68 |
3.08 |
|
P2 |
3.51 |
0.21 |
0.35 |
5 |
2 |
4.53 |
3.12 |
Note: M is the average of explanatory variables; SD1 is the standard deviation of 806 consumers at the individual level. SD2 is the standard deviation obtained by weighing the 70 stores at the overall level.
- Results and Interpretation:
Present the main findings of your analysis. Clearly state the impacts of playfulness, epidemic prevention, convenience benefits, and green logistics on customer satisfaction and loyalty.
Line 425-450
The empirical results found that the most important potential variables affecting consumer loyalty during the COVID-19 epidemic were convenience and green logistics services, and the relationships were positive, with impact coefficients of 0.462 and 0.455, respectively. It meant that in order to avoid the risk of catching the epidemic in groups, consumers prefer free delivery logistics services and efficient shopping services that use AI technology to quickly pick up and deliver goods. In addition, the factors that have the greatest impact on consumer satisfaction are the playfulness effect and the epidemic prevention effect. The influence coefficients are 0.536 and 0.486 respectively, and both have positive relationships. It shows that consumers believe that although it is inconvenient to go out for shopping, they can choose to enjoy shopping through online shopping and online video interaction, taking into account epidemic prevention safety and shopping needs. In the MSEM path relationship diagram, it is found that there are two potential variables that affect customer purchase intention, namely loyalty and satisfaction, and loyalty has a greater effect on purchase intention, with influence coefficients of 0.482 and 0.306 respectively. From the above direct and indirect effects, it is found that because Fresh Hema mainly sells the daily necessities of fresh produce and daily necessities of people's livelihood, during the COVID-19 epidemic prevention period, consumers valued Fresh Hema 's provision of free delivery and customer service under the AI technology the most, which is also the most important factor that affects the consumers' willingness to buy.
Under the new retail industry model, Fresh Hema integrates the online and offline sales model, and if it can provide customers with the most necessary online and offline customer service, it will increase the supply and inventory turnover rate of fresh agricultural products, which not only maintains the quality of agricultural products and avoids waste, but also enhances the one-stop and most efficient service from the production end of the agricultural products to the end of the customer, so as to increase the value of the overall agricultural products supply chain.
- Offer explanations for the observed results based on both theoretical reasoning and the context of Fresh Hema.
The results of this study are also in line with Fresh Hema's current strategy in the face of a rapidly changing market. In addition to the already existing Fresh Hema, it is now actively expanding into Hema Market and Hema Pick'n Go, which provides consumers with greater convenience. Fresh Hema also utilizes a direct-sale supply chain, takes into account both cost and quality, and adopts small-portion packaging to cater to the shopping habits of consumers. In addition, Fresh Hema's continuous upgrading of the consumer scene increases the customer experience and cultivates the formation of new consumer habits.
Fresh Hema has already connected with Taobao and Alipay membership systems to realize data sharing. Using consumers' clear behavioral data, Fresh Hema can carry out advertising and marketing to attract more consumers and form a benign consumption cycle, taking into account both playfulness and epidemic prevention.
- Discussion and Implications:
- Interpret the implications of your findings for the agricultural value chain and the new retail model. Discuss how your results align with the broader trends in consumer behavior and the changing retail landscape.
- Discuss how your results align with the broader trends in consumer behavior and the changing retail landscape.
This phenomenon has had an impact on the agricultural value chain. This study found that Fresh Hema stores had the same prices online and offline. However, fresh product from e-commerce platforms could not provide consumers with a consumption experience comparable to traditional fresh products. Therefore, information regarding product origin, packaging, specifications, and transportation can be disclosed on e-commerce platforms with a page design that attracts consumers' attention. In addition to attracting consumers' attention, page design can make consumers more willing to pay for high-quality fresh food through a comprehensive platform experience and direct audio-visual interaction. This has also created a trend of actively optimizing playfulness through e-commerce platforms selling fresh products. In the post-epidemic era, consumers have got over the fear over the pandemi, and epidemic prevention and safety have been stable. According to the psychology of subsequent compensation, the public will eventually return to the rational new normal; the mastery of price sensitivity of fresh product e-commerce platforms is also a trend that must be addressed in the future.
- Speculate on why convenience benefits and green logistics have a more significant impact on customer loyalty compared to other factors.
According to the empirical analysis results of this study, there are four factors that affect loyalty, namely convenience benefits, green logistics, system services and playfulness. Among them, convenience benefits have the greatest impact on consumer loyalty, with an impact coefficient of 0.462 and a significant T value; The second is that the effect of green logistics services on loyalty is 0.455, and the T value is significant.
According to the analysis of this study, in order to avoid contact between people, which may cause the risk of infection in groups, and to save time for efficient shopping, consumers prefer customer services that have free delivery logistics services and are equipped with AI technology that can quickly pick up goods.
- Comparison with Prior Research:
- Compare your findings with relevant previous studies. Highlight any consistencies or disparities, and discuss potential reasons for these differences.
Regarding the impact of consumer satisfaction on consumer loyalty, this study's findings are consistent with other relevant studies on agriculture (Dardak & Habib, 2010; Sadeli, Utami & Rahmaniss, 2016; Acharya & Lillywhite, 2021; Chaitorn& Saqib, 2022). The findings indicate that under the agricultural value chain, fresh agricultural product e-commerce merchants must improve consumer satisfaction from production and sales to logistics, website management and other technical development aspects. This will further enhance consumer loyalty for fresh agricultural products sold on e-commerce platforms. Furthermore, the difference between this study and previous studies is that the epidemic prevention and safety variables were added to analyze the impact of changes in consumer behavior on consumer satisfaction post-COVID-19; fresh agricultural products e-commerce merchants helped consumers with epidemic prevention through product manufacturing, transportation and sales as the latest, most effective way to enhance brand image and strengthen brand trust and to further improve consumer satisfaction.
- Limitations and Future Research:
Acknowledge the limitations of your study, such as potential biases in self-reported data or limitations specific to the chosen methods. Suggest avenues for future research that could overcome these limitations.
Line 493-509
5. Limitations and Future Research
Only empirical analyses on Fresh Hema were conducted in this study. However, Fresh Hema has more than 300 stores in 27 cities across China. Due to sample limitations, it was not possible to analyze stores in more cities. Specifically, China has a vast territory, and the consumption behavior in first- and second-tier cities may differ due to geographical factors. Therefore, this study only analyzed the available collected store big data, with limitations in making inferences about populations based on samples. Presently, there are various fresh product e-commerce companies in China, including traditional and new fresh food stores. New fresh food stores include home delivery, to store + home delivery, pickup at the drop box, community group purchase, and other models. Different business models of fresh product e-commerce companies meet consumer needs at varying levels. We suggest that in addition to continuously exploring consumer loyalty and satisfaction, fresh product e-commerce companies in China should further explore a sustainable development economic model. Specifically, fresh product e-commerce is not widely used online. Thus, how to create branded and highly differentiated frozen food under the new agricultural value chain to achieve a reasonable gross profit margin is a topic worthy of further research.
- Conclusion:
Summarize the main takeaways from your study. Reiterate the significance of understanding consumer loyalty and satisfaction in the evolving agricultural value chain.
The empirical results show that during the COVID-19 epidemic, the most important potential variable affecting consumer loyalty is the convenience benefit, and it has a positive relationship, with an impact coefficient of 0.462. Said that in order to avoid the risk of cluster infection, consumers prefer efficient shopping with free delivery service and AI technology that can quickly pick up goods. In addition, the factors that have a greater impact on consumer satisfaction are the playfulness effect and the epidemic prevention effect, and both have a positive relationship, indicating that consumers believe that although it is inconvenient to go out for shopping, they can enjoy the fun of online shopping and online video interaction. Achieve the goals of epidemic prevention safety and shopping fun. It is found in the MSEM path relationship diagram that during the COVID-19 epidemic prevention period, consumers pay the most attention to Hema Fresh's free delivery logistics and customer service under AI technology, which are also key factors affecting consumers' purchase intentions.
Under the new retail model, Hema Fresh integrates online and offline physical sales models. If it can provide customers with the most needed online and physical customer services, It will increase the supply and inventory turnover rate of fresh agricultural products. In addition to maintaining the supply quality of agricultural products and avoiding waste, it can also improve the most efficient one-stop service of agricultural products from the production end to the customer end, thereby increasing the overall value of the agricultural product supply chain.
- References:
- Ensure that you provide a comprehensive list of references to support your study. Make sure to include the source from where you obtained the online questionnaire.
The literature has been supplemented.
For the improvement of the methodology, the suggestions are:
- Elaborate on the Method Selection:
Start by providing a rationale for why you chose PCA and MSEM as your chosen methods. Explain how each method addresses specific aspects of your research objectives. Highlight the advantages of using these methods in studying complex consumer behavior phenomena.
The explanatory variables selected for general consumer behavior research and analysis were all subjectively determined based on expert experience. On the surface, the data were logical. However, there may be a high correlation among explanatory variables. This leads to the significant absence of autocorrelation among explanatory variables in the susceptibility statistics in regression analysis, which can further lead to an excessive residual and misjudged estimation result. Therefore, this study used PCA to reduce explanatory variables; the most objective multivariate analysis method was used to extract important explanatory variables with large factor loads. Nine latent variables were defined (see Table 5.1) to avoid a residual that was too large and the estimated T value was not significant or misjudged.
The statistics presented in this study, during COVID-19 in 2022, indicate that among the 70 Fresh Hema stores in Shanghai, China, the variance of consumer purchasing behavior among each store cluster was large and the variance of consumer purchasing behavior in the same store cluster was small due to differing epidemic levels. This indicates that the consumer behavior data of Fresh Hema stores in Shanghai is multilevel and clustered, which is not applicable for using the traditional SEM model. Statistical data has a unique dependency/pairing relationship, which can easily lead to a violation of the hypothesis of sample independence and invalid statistical tests. According to Bentler and Liang (2003), if the sample explanatory variables are multilevel and clustered, Multilevel Structural Equation Modeling (MSEM) must be adopted to solve the above statistical error problem (Bentler & Liang, 2003; Heck & Hox, 2002; Kalpan & Elliott, 1997; Muthén, 1994; Thomas, 2000). Therefore, this study used MSEM to analyze the relationship between consumer behavior and customer satisfaction, and loyalty and purchase intention to determine the accurate operation and marketing strategy of Fresh Hema. The findings can provide the most efficient value supply chain of fresh agricultural products.
This study used principal component analysis (PCA) to define the sum of observed variables of latent variables. This also solved the statistical fallacy caused by the autocorrelation of explanatory variables. However, the impact of measurement errors may also be overlooked, which can lead to a reduction of statistical test power (Bollen, 1989). Therefore, we used CFA to test the underlying factor structure of the contextual variables at the overall level to determine the applicability of the structural model and reduce the limitations of traditional MLM.
References
Bentler, P. M., & Liang, J. (2003). Two-level mean and covariance structure: Maximum Likelihood via an EM algorithm. In S. P. Reise & N. Duan (Eds.), Multilevel modeling: Methodological advances, issues, and applications (pp. 53-70). Mahwah, NJ: Lawrence Erlbaum Associates.
Bollen, K. A. (1989). Structural equation modeling with latent variables. New York: John Wiley
Heck, R. H., & Thomas, S. L. (2000). An introduction to multilevel modeling techniques. Mahwah, NJ: Lawrence Erlbaum Associates.
Muthén, B. O. (1994). Multilevel covariance structure analysis. Sociological Methods and Research, 22, 376-398.
Hox, J. J. (2002). Multilevel analysis: Techniques and applications. Mahwah, NJ: Lawrence Erlbaum Associates.
Kaplan, D., & Elliott, P. R. (1997). A didactic example of multilevel structural equation modeling applicable to the study of organizations. Structural Equation Modeling, 4, 1-24.
- PCA Procedure:
Detail the step-by-step process of PCA. Discuss data preprocessing steps (such as data cleaning, scaling, and normalization) and the specific PCA algorithm you used. Highlight any considerations you took regarding the number of components to retain or the explained variance threshold.
To avoid overlooking possible high autocorrelation among explanatory variables caused by subjective selection of explanatory variables, i.e., the adoption of the so-called repeated explanatory variables resulting in excessive residual and misjudgment of statistical results, we used PCA to explain variable reduction in this study. Specifically, the most objective multivariate analysis method was used to extract explanatory variables with large factor loads. Nine latent variables (without autocorrelation) were derived to avoid the residual being too large and affecting the estimates in the regression analysis, resulting in non-significance and misjudgment.
There were 35 explanatory variables in this study's initial questionnaire. To solve the issue of autocorrelation among explanatory variables, important explanatory variables were objectively selected. Thus, Principal Component Analysis (PCA) was conducted in this study. According to the PCA results, 21 explanatory variables with factor loading greater than 0.5 were retained. Hair et al. suggest that an explanatory variable with factor loading (PCA) below 0.5 should be deleted (Hair et al., 2006). Thus, nine dimension variables were obtained. Finally, the name of the dimension variables was defined according to the explanatory variable with greater contribution to each dimension variable. Thus, the nine dimension variables were green logistics (eigenvalue 4.26), system quality (eigenvalue 3.85), service quality (eigenvalue 43.62), playfulness (eigenvalue 3.72), epidemic prevention (eigenvalue 3.81), service convenience (eigenvalue 3.51), satisfaction (eigenvalue 3.36), loyalty (eigenvalue 3.51) and purchase intention (eigenvalue 3.32).
Table 5.1 shows that the cumulative explanatory variance of the nine dimension variables was 88.65%. This indicates that the dimension variables were highly representative and explanatory. Furthermore, Cronbach's alpha values of the dimension variables were all higher than 0.8, indicating high reliability.
Table 5.1 Principal Component Analysis
|
Questionnaire Variable |
Factor Loading |
Eigenvalue |
Cumulative Explanatory Variance (%) |
Cronbach’s alpha |
|
Dimension Variable 1: Green Logistics |
|
4.26 |
18.5 |
0.92 |
|
G1 Use environmentally friendly packaging |
0.865 |
|
|
|
|
G2 Use environmentally friendly shopping bags |
0.932 |
|
|
|
|
Dimension Variable 2: System quality |
|
3.85 |
33.86 |
0.89 |
|
SQ1 Combine network video systems |
0.931 |
|
|
|
|
SQ2 Provide product information online |
0.886 |
|
|
|
|
SQ3 APP mobile payment service |
0.925 |
|
|
|
|
Dimension Variable 3: Service Quality |
|
3.62 |
46.72 |
0.90 |
|
SEQ1 Combine online and offline services |
0.906 |
|
|
|
|
SEQ2 Limited time delivery service. |
0.871 |
|
|
|
|
SEQ3 Assist in fresh product management services. |
0.936 |
|
|
|
|
Dimension Variable 4: Playfulness |
|
3.52 |
57.68 |
0.88 |
|
L1 Enjoy fresh food service. |
0.895 |
|
|
|
|
L2 Provide parent-child interactive entertainment activities. |
0.902 |
|
|
|
|
Dimension Variable 5: Epidemic Prevention |
|
3.46 |
66.42 |
0.86 |
|
E1 Store personnel take temperature and monitor health. |
0.962 |
|
|
|
|
E2 Epidemic prevention education and training for store personnel. |
0.926 |
|
|
|
|
E3 Regular environment cleaning and disinfection. |
0.906 |
|
|
|
|
Dimension Variable 6: Service Convenience |
|
3.21 |
74.25 |
0.87 |
|
C1 Combine with AI automated pickup and packaging services. |
0.964 |
|
|
|
|
C2 One-package complete shopping service. |
0.922 |
|
|
|
|
Dimension Variable 7: Satisfaction |
|
3.16 |
80.98 |
0.85 |
|
S1 Customer satisfaction with fresh products. |
0.891 |
|
|
|
|
S2 Customer's overall satisfaction with store service. |
0.906 |
|
|
|
|
Dimension Variable 8: Loyalty |
|
3.11 |
85.27 |
0.83 |
|
LO1 Customers will continue to buy products in the store. |
0.937 |
|
|
|
|
LO2 Customers will continue to recommend friends and family members to the store. |
0.928 |
|
|
|
|
Dimension Variable 9: Purchase Intention |
|
3.02 |
88.65 |
0.84 |
|
P1 Customers are willing to buy products in the store. |
0.896 |
|
|
|
|
P2 Customers spend more time and buy more products in the store. |
0.901 |
|
|
|
|
Total cumulative explanatory variance (%) |
|
|
88.65 |
|
- Interpretation of PCA Components:
- After applying PCA, provide a detailed interpretation of the identified principal components. Explain how these components capture the essential factors of online consumer behavior. Relate these components back to the theoretical constructs in your research.
Based on the PCA results, this study used six of the nine dimension variables, i.e., green logistics, system quality, service quality, playfulness, epidemic prevention, and service convenience, as latent independent variables for MSEM analysis. Moreover, three of the nine dimension variables, i.e., satisfaction, loyalty, and purchase intention were used as latent dependent variables to explore the causal path relationship between the latent independent variables of Fresh Hema's customer service and the latent dependent variables of consumers (Anderson and Gerbing, 1988).
According to the comparative analysis results of the three models of MSEM, the M3 model has the best configuration. In the M3 model, the main variables that affect Loyalty are Green Logistics, System quality, Playfulness, and Service convenience. The main variables that affect Satisfaction are System quality, Service quality, Playfulness and Epidemic prevention. Indirect influences on consumer Purchase Intention include Satisfaction and Loyalty.
Reference
Anderson, J. C., & Gerbing, D. W. (1988). Structural equation modeling in practice: A review and recommended two-step approach. Psychological Bulletin, 103, 411-423.
- MSEM Model Specification:
- Describe the structural equation model you constructed in more detail. Elaborate on the theoretical underpinnings of the variables included in the model. Explain why certain variables were chosen and how they relate to the concepts of customer loyalty and satisfaction.
Among the nine latent variables obtained through PCA, there was no regression analysis fallacy caused by autocorrelation. Therefore, when the causal path relationship between latent independent variables and latent dependent variables was conducted in this study, correct and objective results could be obtained.
This study analyzed the factor structure at the individual level (within the cluster) and the overall level (between the clusters) simultaneously. At the individual level, we adopted the 9-factor oblique model obtained using PCA. At the overall level, there were 21 observed variables. The first model, M1, was a fully independent multi-level model, which indicates that the overall level had no special structure. This model did not have any assumed factor structure, and served as a baseline model. The factor structure of the overall level can be used by the model setting of the individual level. M2 was the second model, which was partially independent; inter-cluster factors were not correlated. Finally, M3 was the structural model. (Partially independent model with partial correlation among factors between groups . Abbreviation: structural model).
In the analysis of consumer behavior, the main purpose of customer service provided by Fresh Hema during the epidemic period was to satisfy customers, enhance customer loyalty and increase purchase intention. Among the above-mentioned three models, M3 had the best applicability (see Table 5.2). Therefore, this study adopted M3 and applied six customer service items, i.e., green logistics, system quality, service quality, playfulness, epidemic prevention, and service convenience; and three Fresh Hema business objectives, i.e., satisfaction, loyalty, and purchase intention, to analyze the causal path relationship between the latent independent variables of customer service and the latent dependent variables of the business objectives of Fresh Hema.
- MSEM Estimation and Fit Measures:
- Provide a comprehensive overview of the estimation process in MSEM. Explain the statistical software you used and any relevant syntax or commands. Discuss how you assessed the model fit using various fit indices (e.g., CFI, RMSEA) and provide cutoff criteria for determining acceptable fit.
This study used Mplus4 (Muthén & Muthén, 2004) for MSEM analysis. The advantage of this method is that it can directly estimate the sample covariant matrix. Furthermore, it does not need other software or a calculation program to obtain SW and SB. The Mplus syntax of the MSEM analysis used in this study is attached (see Attachment 1).
This study performed a model fit measure among the above three multi-level models, and the results are shown in Table 5.2. We found that M1 did not have any setting of factor structure; RMSEA is 0.15, higher than M3’s 0.028, M1’s CFI is 0.71, lower than M3’s 0.92. thus, the mode fit was the least ideal. M3 had the best fit and the best model. Therefore, this study used M3 to explore the causal path relationship between the latent independent variables of customer service provided by Fresh Hema (green logistics, system quality, service quality, playfulness, epidemic prevention, service convenience) and the latent dependent variables of consumers (satisfaction, loyalty, and purchase intention).
Table 5.2 Multi-level MSEM Model Fit Index Table
|
Model
|
χ2/df |
RMSEA |
CFI |
TLI |
SRMR |
||
|
Intra-cluster |
Inter-cluster |
||||||
|
M1 |
Independent model (no correlation between all factors) |
5.38 |
0.150 |
0.71 |
0.68 |
0.62 |
0.44 |
|
M2 |
Partially independent model (no correlation between the inter-cluster factors) |
3.15 |
0.076 |
0.83 |
0.91 |
0.43 |
0.42 |
|
M3 |
Structural model |
2.36 |
0.052 |
0.92 |
0.96 |
0.32 |
0.38 |
- Handling of Missing Data and Assumptions:
- Address the issue of missing data if applicable. Explain your approach for handling missing values and its potential impact on the results. Discuss any assumptions of MSEM (e.g., normality, linearity) and how you assessed their validity.
According to the impact of missing data on this study's analysis results, the greater the missing ratio, the greater the difference between the variance matrix estimated from the missing data and the variance matrix calculated from the baseline data set in the "modified likelihood ratio statistic." This study analyzed MSEM using a linear regression model. Therefore, the maximum likelihood approximation method was used to process missing data and reduce the variance matrix as much as possible.
- Results Interpretation:
Expand upon the interpretation of your MSEM results. Discuss the magnitude, direction, and significance of the relationships between the selected variables. Relate these findings back to your research hypotheses and the theoretical framework.
The purpose of this study is to analyze the relationship paths of Hema Fresh’s six main customer services to consumer satisfaction, loyalty and purchase intention under the new retail model (integrating online and physical).
During 2022, mainland China has been severely impacted by COVID-19. In addition to affecting consumers' health and eating habits, it has also changed consumers' purchasing motivations and behaviors for agricultural products, fresh food and other livelihood necessities. Therefore, this study conducted 11 hypothesis tests based on the MSEM relationship path. The results were all in line with the hypotheses, and the T values ​​were all significant at the significance level α=0.05.
According to the empirical results, the important factor affecting consumers' loyalty to Hema Fresh is convenience, and it has a positive relationship, with an impact coefficient of 0.462. Observing the actual environment and research results show that in order to avoid the risk of cluster infection, consumers prefer efficient shopping with free delivery logistics services and AI technology that can quickly pick up goods.
During the epidemic prevention period, the major factors affecting consumers' satisfaction with the retail industry are the playfulness effect and the epidemic prevention effect, and both of them are positively related, indicating that consumers believe that during the epidemic prevention period, consumers not only enjoy the fun of online shopping, but also attach importance to epidemic prevention completeness of the measure. It is also found in the MSEM path relationship diagram that customer loyalty has a greater impact on consumers' willingness to purchase in the retail industry than customer satisfaction, and the impact coefficients are 0.482 and 0.306, respectively. Based on the above, because Hema Fresh mainly sells fresh agricultural products and daily necessities that are necessary for people’s daily lives, these supplies are even more important during the COVID-19 epidemic prevention period. Therefore, consumers attach the most importance to Hema Fresh’s provision Free delivery and efficient shopping with AI technology services are also the main services that influence consumers’ purchasing intentions.
- Model Comparison and Sensitivity Analysis:
Consider conducting sensitivity analyses or model comparisons. This could involve comparing different models with varying sets of variables or testing alternative specifications. Discuss how these additional analyses support the robustness of your findings.
In order to obtain the best empirical model, this study compares the suitability of three multi-level models.M1 is the overall level and has no special structure settings; M2 is a partially independent model, and there is no correlation between factors between groups; M3 is a partially independent model, and there is a partial correlation between factors between groups (abbreviation: structural model).
According to the results in Table 5.2 above, it is found that when the M1 overall hierarchical structure has no specific settings at all, the fit of the model is the worst, with RMSEA of 0.150 and CFI of 0.72. From the inter-group SRMR index (0.44), it can be seen that the standardized residuals of the overall level It is very large, and the residual at the individual level is relatively small (SRMR=0.62), indicating that the poor fit of M1 mainly occurs at the overall level. On the contrary, for the M3 model, the RMSEA dropped to 0.028, the CFI increased to 0.92, and the model configuration was moderately improved. It is the most representative multi-layer structure model with the best configuration among the three models.
- Discussion of Implications:
Extend the discussion of your findings to include the practical implications of the results. How can the identified factors impact consumer behavior and the performance of the new agricultural value chain? Discuss how businesses or policymakers could leverage these insights.
According to the empirical results of this study, it is found that under the impact of the COVID-19 epidemic and changes in the living environment, consumers prefer time-saving and efficient consumption methods. Fresh agricultural products are the necessities of consumers' livelihood and have a stable demand. However, agricultural products are not easy to store and cannot be stored for a long time. Due to sales channel problems, agricultural products are often damaged and wasted, disrupting the supply value chain of agricultural products, and affecting the balance of supply and demand in the agricultural product market.
Under the new retail industry model, Hema Fresh integrates online and offline sales models to provide online and offline customer services, which can not only meet consumers' COVID-19 epidemic prevention safety, convenience and efficient consumption patterns, but also Increasing the supply turnover rate and inventory turnover rate of fresh agricultural products can not only maintain the quality of agricultural products and avoid waste, but also improve the one-stop and most efficient service of agricultural products from the production end to the customer end, so as to increase the overall value supply chain of agricultural products .
- Limitations and Future Research:
Clearly outline the limitations of your methodology. Discuss potential sources of bias, limitations of the chosen methods, and constraints in generalizing the results. Suggest avenues for future research that could build upon or address these limitations.
Principal component analysis (PCA) was used to define the sum of observed variables of the latent variables, which also solved the statistical fallacy caused by the autocorrelation of explanatory variables. However, the impact of measurement errors may be overlooked, which can lead to a reduction of statistical test power (Bollen, 1989). Therefore, this study used CFA to test the applicability of the underlying factor structure of the contextual variables at the overall level. This method addresses the subjective limitations of traditional MLM and applies the underlying construct methodology to explore contextual effects. This may be extended to more academic issues in the future.
Although there are many advantages in applying the MSEM model to test latent variables, if there are many estimated parameters and the model is divided into individual and overall levels, the stability and convergence of parameter estimation will be greatly challenged.
Reference
Bollen, K. A. (1989). Structural equation modeling with latent variables. New York: John Wiley
- Conclusion of Methodology:
- Summarize the strength and appropriateness of using PCA and MSEM for your study. Reiterate how the combined use of these methods allowed you to comprehensively investigate the impact of consumer loyalty and satisfaction in the new agricultural value chain.
Based on the above, this study uses PCA to reduce explanatory variables, extracts important explanatory variables using objective statistical methods, and defines 9 potential variables (as shown in Table 5.1). At the same time, it avoids self-correlated collinearity of explanatory variables, which causes problems in regression analysis. The residual term is too large, which leads to insignificant T value and misjudgment.
According to narrative statistics, it was found that during the COVID-19 epidemic in 2022, various regions adopted epidemic isolation measures, resulting in differences in consumer behavior among the 70 Hema Fresh stores in Shanghai in mainland China, resulting in consumption between store groups. The variation of consumer purchasing behavior is very large, and the variation of consumer purchasing behavior within the branch store group is very small, indicating that consumer behavior has multi-level data and clustered characteristics, and the statistical data has a dependent/dual relationship, which can easily lead to violations. The sample independence assumption and statistical test are invalid, and the traditional SEM model is not applicable. Therefore, the multilevel analysis technique (MSEM) is used to solve the above statistical error problem (Bentler & Liang, 2003; Heck & Thomas, 2000; Muthén, 1994; Hox, 2002; Kalpan & Elliott, 1997).
In order to find the best MSEM factor structure model, the suitability of three multi-level models was compared, and it was found that the M3 structural model had an RMSEA of 0.028 and a CFI of 0.92, which was the best configuration and the most representative multi-level structure model.
This study uses the M3 model to analyze the causal path relationship between Hema Fresh’s customer service under the new retail model (integrating online and physical) on consumer satisfaction, loyalty and purchase intention.
Empirical results found that during the COVID-19 epidemic, the most important potential variable affecting consumer loyalty was convenience, with a positive relationship and an impact coefficient of 0.462. Said that in order to avoid the risk of cluster infection, consumers prefer shopping services with free delivery services and AI technology that can quickly pick up goods. In addition, the major factors affecting consumer satisfaction are the playfulness effect and the epidemic prevention effect, and both are positively related, indicating that consumers believe that although it is inconvenient to go shopping, they can choose the fun of online shopping and online video interaction, Taking into account both epidemic prevention safety and shopping fun. According to the empirical results of the MSEM path relationship diagram, during the COVID-19 epidemic prevention period, consumers pay the most attention to Hema Fresh's free delivery and customer service under AI technology, which are also key factors affecting consumers' willingness to purchase.
Under the new retail industry model, Hema Fresh integrates online and offline physical sales models. If it can provide customers with the most needed online and physical customer services, it will increase the supply turnover rate and inventory turnover of fresh agricultural products. In addition to maintaining the supply quality of agricultural products and avoiding waste, it can also improve the most efficient one-stop service of agricultural products from the production end to the customer end, thereby increasing the overall agricultural product value supply chain.
Comments on the Quality of English Language The suggestions are:
- Language and Clarity: Ensure that the language used is clear, precise, and easily understandable. Avoid unnecessary jargon and technical terms that might confuse readers.
p.112 Already corrected shopping malls to online fresh food shopping malls.
- Tables and Figures: If relevant, consider including tables or figures to present the data in a visual and organized manner, making it easier for readers to comprehend the findings.
We provide the relevant charts in the modification. For example, the following chart:
Table 5.1 Principal Component Analysis
Table 5.2 Multi-level MSEM Model Fit Index Table
Table 6.1: Descriptive Statistical Analysis of the 21 Variables regarding Consumer Behavior of Fresh Hema Stores
- Formatting and Citations: Follow the formatting guidelines of the target journal and ensure that all sources are properly cited using the appropriate citation style.
All sources included in this study were properly cited using appropriate citation style.
- Grammar and Proofreading: Thoroughly proofread the article to correct any grammatical errors or typos. Seek feedback from colleagues or peers to ensure the clarity and quality of the writing.
The content of this research has been translated and corrected by English majors.
Appendix 1: The Mplus syntax of this study (taking the M3 model as an example)
TITLE: 2level MLSEM ANALYSIS (M3)
DATA: FILE IS hm3.dat;
VARIABLE: NAMES ARE G1-G2 SQ1-SQ3 SEQ1-SEQ3 L1-L2 E1-E3 C1-C2 S1-S2 LO1-LO2 P1-P2 clus;
CLUSTER = clus;
ANALYSIS: TYPE IS TWOLEVEL;
MODEL:
%WITHIN%
Green1 BY G1-G2;
Syste1 BY SQ1-SQ3;
Serv1 BY SEQ1-SEQ3;
Play1 BY L1-L2;
Epid1 BY E1-E3;
Conve1 BY C1-C2;
Satis1 BY S1-S2;
Loy1 BY LO1-LO2;
Purc1 BY P1-P2;
Satis1 on Syste1 Serv1 Epid1 Conve1 ;
Loy1 on Green1 Syste1 Play1 Conve1;
Purc1 on Satis1 Loy1;
%BETWEEN%
Green2 BY G1-G2;
Syste2 BY SQ1-SQ3;
Serv2 BY SEQ1-SEQ3;
Play2 BY L1-L2;
Epid2 BY E1-E3;
Conve2 BY C1-C2;
Rating2 BY R1-R3;
Satis2 BY S1-S2;
Loy2 BY LO1-LO2;
Purc2 BY P1-P2;
Satis2 on Syste2 Serv1 Epid2 Conve2 ;
Loy2 on Green2 Syste2 Play2 Conve2;
Purc2 on Satis2 Loy2;
OUTPUT:
SAMPSTAT;
STANDARDIZED;
CINTERVAL;

Reviewer 3 Report
The topic of the paper – the impact of consumer loyalty and customer satisfaction in new agricultural value chain - is relevant and very interesting. Authors used a wide range of international literature sources and cited them correctly. Most part of them are from the last few years, but I suggest them to add more adequate sources. I think that the Authors formulated too much - 11 – hypotheses, several of them can be evident. Probably that was the reason that all of them were accepted. In my mind, this means that the Authors knew everything, so there is nothing new in their paper. So, I suggest reducing the number and reformulate them. I agree the methodological part.
Author Response
Reviewers’ Comments and Suggestions for Author (Round 1)
5 September 2023
Dear Reviewer,
Thank you for the constructive suggestions and comments on our manuscript(ID: agriculture-2574699). The suggestions and comments are helpful for improving the manuscript. We are submitting the revised version of the manuscript with our responses to the suggestions and comments by the reviewer. Many thanks for your guidance.
Our responses to each suggestion and comment are as follows, and they are presented in blue texts with a grey background color in the revised manuscript.
The topic of the paper – the impact of consumer loyalty and customer satisfaction in new agricultural value chain - is relevant and very interesting. Authors used a wide range of international literature sources and cited them correctly. Most part of them are from the last few years, but I suggest them to add more adequate sources. I think that the Authors formulated too much - 11 – hypotheses, several of them can be evident. Probably that was the reason that all of them were accepted. In my mind, this means that the Authors knew everything, so there is nothing new in their paper. So, I suggest reducing the number and reformulate them. I agree the methodological part.
Response:
Thank you very much for your comments and suggestion. The modifications are as follows:
- I suggest them to add more adequate sources.
Here are more literature on consumer behavior in the context of agricultural value chains, new retail models, and the role of AI and environmental awareness. Yan et al. (2020) reported that the fresh agricultural product supply chain's circulation efficiency is significantly influenced by the purchasing power of end consumers. Consumers are an essential factor in determining agricultural value chains. Mowat and Collins (2000) found that supply chains in new and emerging agricultural industries typically lack the ability to link product quality with consumer behaviour.
Agricultural value chains have also changed over time. Dong (2021) found that agricultural value chains are going through a paradigm shift from efficiency-driven to resilience-focused eco-friendly agriculture. Other factors affecting agriculture value chains include AI. Ganeshkumar et al. (2023) indicated that with AI adoption in agriculture, value chains can increase agricultural income, enhance competitiveness, and reduce cost. Among the agriculture value chains stages, AI research related to agricultural processing and consumer sector is limited compared to input, production, and quality testing. Based on the above relevant literature, it is evident that consumer behavior plays an essential role in agriculture value chains. The present study focuses on the increasing trend of consumers' demand for fresh agricultural products after COVID-19, addressing the gap in previous studies that primarily focused on general agricultural products (Feldmann & Hamm, 2015; Dimitri, Oberholtzer & Pressman, 2016; Grebitus, Printezis & Printezis, 2017). This study also examines which parties are most suited to facilitate value chain upgrading.
- I suggest reducing the number and reformulate them
Thank you for your suggestion. The author formulated these 11 research hypotheses based on the relevant literature. Given China's strong domestic market demand for fresh agricultural products and the rapid development of artificial intelligence, the author plans to explore whether consumer behavior differs. Meanwhile, this study contributes to the literature by adding epidemic prevention and safety variables that have not been previously examined in relevant studies, which further conform to the current development status worldwide. Since the empirical analysis of this study has been completed, the author will follow your valuable suggestions to formulate more precise research hypotheses in future research.
Moreover, the MSEM model in this study can be simplified by reducing the number of variables and hypothesis to avoid the estimation errors caused by the failure of data convergence. Fortunately, the estimates of the data from the 11 hypothesis in this study are convergent and the t-values are significant.

Round 2
Reviewer 2 Report
The content has been improved, but the charts can still be enhanced for aesthetic and neatness.
The sentences can also be further verified for accuracy and logic.
Author Response
Reviewers’ Comments and Suggestions for Author (Round 2)
10 September 2023
Dear Reviewer,
Thank you for the constructive suggestions and comments on our manuscript(ID: agriculture-2574699). The suggestions and comments are helpful for improving the manuscript. We are submitting the revised version of the manuscript with our responses to the suggestions and comments by the reviewer. Many thanks for your guidance.
Our responses to each suggestion and comment are as follows, and they are presented in red texts with a grey background color in the revised manuscript.
Comments and Suggestions for Authors
The content has been improved, but the charts can still be enhanced for aesthetic and neatness.
Response:
Thank you very much for your comments and suggestion. The modifications are as follows:
Line178-180
Since enhancing customer satisfaction is an important strategy to change consumer behavior and build consumer loyalty, it has become the focus of the industry.
Line208-211
Customer loyalty refers to the continuous emotional relationship between a company or organization and its customers, which is manifested in the customers' repeated purchasing and continuous interaction with the company, and is an integral part of the company's growth and development.
Line341
Table 1. Factor Analysis of the Dimension Variables.
|
1. Green Logistics |
G1 Use environmentally friendly packaging. |
|
G2 Use environmentally friendly shopping bags. |
|
|
2. System Quality |
SQ1 Combine with network video system |
|
SQ2 Provide all product information on the network. |
|
|
SQ3 APP mobile payment service. |
|
|
3. Service Quality |
SEQ1 Combine online and offline services (Online and Offline). |
|
SEQ2 Limited time delivery service. |
|
|
SEQ3 Assist in fresh product management services. |
|
|
4. Playfulness |
L1 Enjoy fresh food service. |
|
L2 Provide parent-child interactive entertainment activities. |
|
|
5. Epidemic Prevention |
E1 Store personnel take temperature and monitor health. |
|
E2 Epidemic prevention education and training for store personnel. |
|
|
E3 Regular environment cleaning and disinfection. |
|
|
6. Service Convenience |
C1 Combine with AI automated pick-up and packaging services. |
|
C2 One-package complete shopping service. |
|
|
7. Satisfaction |
S1 Customer satisfaction with fresh products. |
|
S2 Customer's overall satisfaction with store service. |
|
|
8. Loyalty |
LO1 Customers will continue to buy products in the store. |
|
LO2 Customers will continue to recommend friends and family members to the store. |
|
|
9. Purchase Intention |
P1 Customers are willing to buy products in the store. |
|
P2 Customers spend more time and buy more products in the store. |
Line359
Table 2. Measurement Model Goodness-of-fit Results.
|
Goodness-of-fit Index |
Threshold Value |
Measured Value |
Result Determination |
|
Ratio of X2 to degrees of freedom (X2 / DF ) |
≤ 3.00 |
2.22 |
Acceptable |
|
Goodness-of-fit index (GFI) |
≥ 0.80 |
0.95 |
Acceptable |
|
Adjusted Goodness-of-fit index (AGFI) |
≥ 0.80 |
0.92 |
Acceptable |
|
Normed fit index (NFI) |
≥ 0.90 |
0.96 |
Acceptable |
|
Comparative fit index (CFI) |
≥ 0.90 |
0.93 |
Acceptable |
|
Root Mean Square Residual (RMSR) |
≤ 0.05 |
0.036 |
Acceptable |
Line371
Table 3. Reliability Analysis.
|
Dimension Variable |
Standardized Factor Loading |
Standard Error (SE) |
t Value |
Composite Reliability (CR) |
Average Variance Extracted (AVE) |
|
1. Green Logistics G1 G2 |
0.865 0.932 |
0.021 0.034 |
15.36 12.32 |
0.936 |
0.786 |
|
2. System quality SQ1 SQ2 SQ3 |
0.931 0.886 0.925 |
0.023 0.012 0.028 |
14.62 16.32 13.75 |
0.925 |
0.875 |
|
3. Service quality SEQ1 SEQ2 SEQ3 |
0.906 0.871 0.936 |
0.028 0.035 0.021 |
12.36 15.21 14.31 |
0.956 |
0.902 |
|
4. Playfulness L1 L2 |
0.895 0.902 |
0.031 0.024 |
14.36 11.32 |
0.928 |
0.879 |
|
5.Epidemic prevention E1 E2 E3 |
0.962 0.926 0.906 |
0.016 0.027 0.031 |
15.32 12.68 13.71 |
0.938 |
0.901 |
|
6.Service convenience C1 C2 |
0.964 0.922 |
0.028 0.021 |
14.38 11.62 |
0.918 |
0.906 |
|
7. Satisfaction S1 S2 |
0.891 0.906 |
0.025 0.019 |
9.77 12.36 |
0.941 |
0.865 |
|
8. Loyalty LO1 LO2 |
0.937 0.928 |
0.031 0.028 |
11.28 13.25 |
0.928 |
0.906 |
|
9. Purchase Intention P1 P2 |
0.896 0.901 |
0.014 0.029 |
12.27 13.21 |
0.912 |
0.892 |
Line381
Table 4. Structural Model Goodness-of-fit Results.
|
Goodness-of-fit Index |
Threshold Value |
Measured Value |
Result Determination |
|
Ratio of X2 to degrees of freedom (X2 / DF ) |
≤ 3.00 |
2.36 |
Acceptable |
|
Goodness-of-fit index (GFI) |
≥ 0.80 |
0.93 |
Acceptable |
|
Adjusted Goodness-of-fit index (AGFI) |
≥ 0.80 |
0.91 |
Acceptable |
|
Normed fit index (NFI) |
≥ 0.90 |
0.95 |
Acceptable |
|
Comparative fit index (CFI) |
≥ 0.90 |
0.92 |
Acceptable |
|
Root Mean Square Residual (RMSR) |
≤ 0.05 |
0.028 |
Acceptable |
Line388
*** indicates p< 0.01; ** indicates p< 0.05
Figure 2. Estimate Results of the Path Coefficient of the Research Model.
Line458-485
This study investigates the effects of green logistics, system quality, service quality, playfulness, safety and epidemic prevention, convenience benefits, customer satisfaction, and loyalty on purchase intention of Alibaba's Fresh Hema in Mainland China. The hypotheses proposed in this study are all supported by the results of structural equation modeling. Among them, the variable of playfulness has the greatest positive effect on customer satisfaction, followed by the variable of epidemic prevention and safety, which indicates that consumer behavior has changed due to the increasing development of smart technology and the impact of the epidemic. In other words, from the point of view of playfulness, consumers tend to be attracted to interesting and joyful things. Therefore, it is recommended that online fresh food shopping malls adopt game-based marketing as one of the feasible strategies in the future. For example, online fresh food shopping malls can grasp the principle of "big prizes and lots of small prizes" by means of bonus point rewards, cash rebates for spending, friend referral gifts, achievement medals, or leaderboard competitions, etc., so as to make online shopping more fun, thereby increasing the degree of interaction between the mall and its customers, and further enhancing customer satisfaction. In addition, as AI technology allows businesses to further integrate physical and online stores, providing a seamless online mall shopping experience for agricultural products is also a viable direction for the future. For example, the recommendation engine driven by AI technology can accurately pinpoint each consumer's preference for agricultural products and even eating habits, and further generate more personalized recommendation content when consumers browse the mall, so that the appropriate products can be recommended to the most suitable consumers in the shortest possible time.
In addition, among the potential variables affecting customer loyalty, the ones with the greatest effect are the convenience benefit and the green logistics, indicating that consumers place the greatest importance on the convenience benefit of shopping as well as green awareness in their busy lives. Therefore, we recommend that the online fresh food mall can strengthen the service function of customers in the front office so that consumers can improve the convenience and safety of using the mall.
Line492-494
Regarding the back office of the online fresh food mall, the return and exchange mechanism, payment security, and logistics distribution should be adequately planned to reduce the unpredictability of consumers searching for agricultural products.
Comments on the Quality of English Language
The sentences can also be further verified for accuracy and logic.
Here are some proofs that our articles have been proofread by professionals.
